# Insights into beta cell regeneration for diabetes via integration of molecular landscapes in human insulinomas

Huan Wang[1,2,14], Aaron Bender[2,3], Peng Wang[3], Esra Karakose[3], William B. Inabnet[4], Steven K. Libutti[5], Andrew Arnold[6], Luca Lambertini[7], Micheal Stang[8], Herbert Chen[9], Yumi Kasai[10], Milind Mahajan[1], Yayoi Kinoshita[11], Gustavo Fernandez-Ranvier[4], Thomas C. Becker[12], Karen K. Takane [3], Laura A. Walker[3], Shira Saul [3], Rong Chen[1,14], Donald K. Scott[3], Jorge Ferrer [13], Yevgeniy Antipin[1,14], Michael Donovan[11], Andrew V. Uzilov[1,14], Boris Reva[1], Eric E. Schadt [1,14], Bojan Losic[1], Carmen Argmann[1] & Andrew F. Stewart[3]

Although diabetes results in part from a deficiency of normal pancreatic beta cells, inducing human beta cells to regenerate is difficult. Reasoning that insulinomas hold the "genomic recipe" for beta cell expansion, we surveyed 38 human insulinomas to obtain insights into therapeutic pathways for beta cell regeneration. An integrative analysis of whole-exome and RNA-sequencing data was employed to extensively characterize the genomic and molecular landscape of insulinomas relative to normal beta cells. Here, we show at the pathway level that the majority of the insulinomas display mutations, copy number variants and/or dys-regulation of epigenetic modifying genes, most prominently in the polycomb and trithorax families. Importantly, these processes are coupled to co-expression network modules associated with cell proliferation, revealing candidates for inducing beta cell regeneration. Validation of key computational predictions supports the concept that understanding the molecular complexity of insulinoma may be a valuable approach to diabetes drug discovery.

[1] The Department of Genetics and Genomic Sciences and The Icahn Institute for Genomics and Multiscale Biology, The Icahn School of Medicine at Mount Sinai, New York, NY 10029, USA. [2] The Graduate School, The Icahn School of Medicine at Mount Sinai, New York, NY 10029, USA. [3] The Diabetes Obesity and Metabolism Institute, The Icahn School of Medicine at Mount Sinai, New York, NY 10029, USA. [4] The Department of Surgery, The Icahn School of Medicine at Mount Sinai, New York, NY 10029, USA. [5] The Cancer Institute of New Jersey, Rutgers University, New Brunswick, NJ 08901, USA. [6] Center for Molecular Medicine, University of Connecticut School of Medicine, Farmington, CT 06030, USA. [7] The Departments of Environmental Medicine and Public Health and Obstetrics, Gynecology, and Reproductive Sciences, The Icahn School of Medicine at Mount Sinai, New York, NY 10029, USA. [8] The Department of Surgery, Duke University School of Medicine, Durham, NC 27710, USA. [9] The Department of Surgery, University of Alabama at Birmingham, Birmingham, AL 35233, USA. [10] The New York Genome Center, New York, NY 10013, USA. [11] The Department of Pathology, The Icahn School of Medicine at Mount Sinai, New York, NY 10029, USA. [12] The Sarah W. Stedman Center for Nutrition and Metabolism, Duke University School of Medicine, Durham, NC 27710, USA. [13] The Department of Genetics in Medicine, Imperial College, London, W12 0NN, UK. [14]Present address: Sema4, a Mount Sinai venture, Stamford, CT 06902, USA. Huan Wang, Aaron Bender, Peng Wang, Esra Karakose, William B. Inabnet, and Steven K. Libutti contributed equally to this work. Eric E. Schadt, Bojan Losic, Carmen Argmann, and Andrew F. Stewart jointly supervised this work. Correspondence and requests for materials should be addressed to A.F.S. (email: andrew.stewart@mssm.edu)

Normal physiologic human beta cell replication occurs only transiently in human infancy and early childhood, ceasing irreversibly thereafter[1]. Therapeutically, there is only one class of drugs, still in early development, that reproducibly induces human beta cell replication: the harmine analogue class of small molecules that inhibit the kinase, DYRK1A[2–4]. Even here, however, the replication rates induced are modest and not beta cell-specific. Accordingly, there is an urgent need to discover additional beta cell mitogenic drugs and regenerative pathways.

Insulinomas are very rare, small (~ 1–2 cm), slowly proliferating pancreatic beta cell adenomas[5, 6]. They come to medical attention through their overproduction of insulin, causing hypoglycemia, with resultant psychomotor symptoms[5, 6]. They are almost always benign, and are readily treated by laparoscopic removal. Since they are a rare tumor, they are not captured in large cancer genomic surveys such as The Cancer Genome Atlas (TGCA) or the International Cancer Genome Consortium (ICGC).

Here we report whole-exome sequencing (WES) and RNA sequencing (RNAseq) of thirty-eight human insulinomas. We provide these findings for public access with extensive sets of annotations relating to the DNA variants identified, with the ability to prioritize selection of high-impact mutations in a user-defined way.

Our primary intent was to employ an integrative genomics approach to identify mitogenic mechanisms with potential application for human beta cell expansion (Supplementary Fig. 1). This approach entails integrating whole-exome and RNA-sequencing data into network analysis to computationally model insulinoma molecular events relative to normal adult and juvenile human beta cells. We reasoned that although some molecular events in insulinoma are likely relevant to the mechanisms of tumor formation, some may serve to uncover the genetic mechanisms that enforce beta cell quiescence, and are bypassed in such benign tumors. We further validated combinations of lead candidate genes derived from this approach as beta cell mitogenic mediators. Notably, we focused on insulinomas from subjects not known to be members of multiple endocrine neoplasia type 1 (MEN1) kindreds, as the MEN1 gene has been previously reported as one of the most frequently mutated genes in hereditary pancreatic neuroendocrine tumors (PNETs), although MEN1 mutations are uncommon in sporadic insulinomas[5–7]. Despite attempting to exclude MEN1 subjects, we nevertheless find widespread abnormalities in genes functionally related to MEN1, revealing a previously unsuspected unifying mechanism underlying insulinoma.

## Results

**Insulinomas harbor recurrent mutations in epigenetic genes.** WES was performed on genomic DNA from insulinomas and patient-matched blood cells (as normal controls) from 22 patients, with a mean usable sequencing depth of 79X and 105X for the blood and tumor samples, respectively. We included WES data from four additional insulinomas from a prior report[8], yielding a total of 26 insulinomas with paired normal and tumor DNA data (Supplementary Tables 1, Supplementary Data 1). Mutation analysis was performed as described[9, 10]. After manual

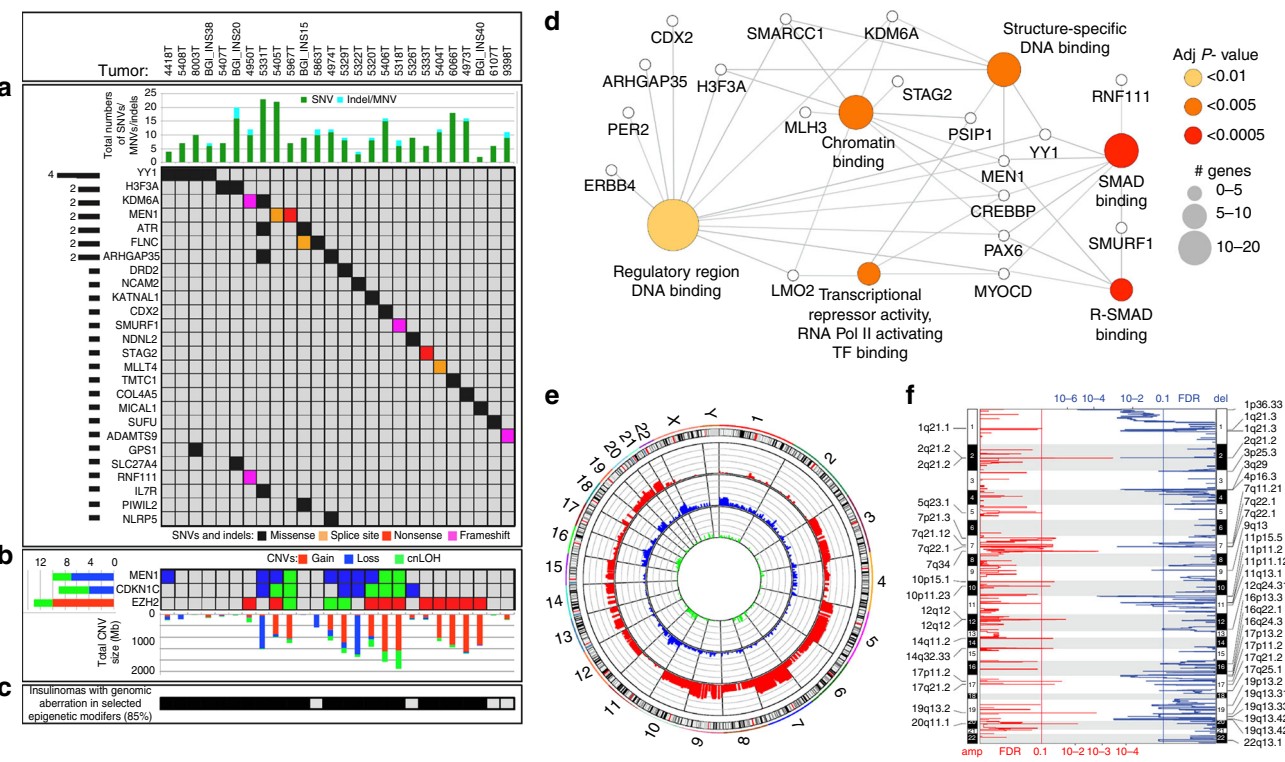

**Fig. 1** Global molecular characterization of the insulinoma genomic landscape from whole-exome sequencing data. **a** A summary of 26 insulinomas (*top box*) and a subset of their protein-changing key driver variants, each sample displaying the variant with the highest allelic fraction and, if any, the variant from recurrently mutated genes. All variants are somatic, except for a germline *MEN1* variant from sample 5967T (at chr11:64,572,613, G>A, p.R420*, nonsense). **b** A summary of somatic copy number variants from selected model-predicted epigenetic modifiers and *CDKN1C*. **c** A summary showing which 85% of insulinomas harbor mutations or CNVs in selected epigenetic modifiers. **d** GO pathway analysis (Molecular Function) of insulinoma key driver variants reveals terms associated with chromatin-binding and SMAD signaling as the most prominent pathways. **e** Circos plot of copy number gain (*red*), loss (*blue*), or copy-neutral cnLOH (*green*). Each *line* within a track represents 20% of the total number of insulinomas. Note that ~ 20% of insulinomas have CNV loss on chromosome 11 (*blue*), and ~ 20–40% have CNV gain on chromosome 7 (*red*). **f** GISTIC2.0 analysis showing finer mapping of regions of significant chromosomal amplification or deletion throughout the genome. *Cytoband labels* indicate significant calls (FDR < 0.1)

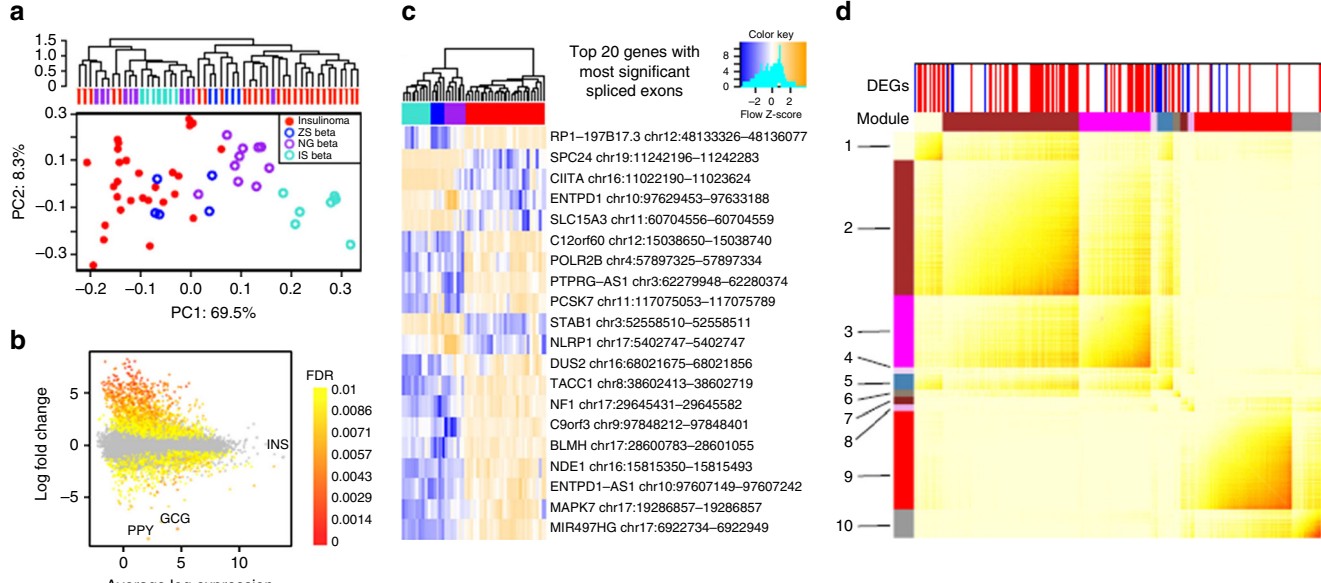

**Fig. 2** Global molecular characterization of transcriptomes from the 25 insulinomas compared with beta cells from 22 normal subjects. **a** Unsupervised classification analysis including hierarchical clustering (*above*) and principal component analysis (*below*) of insulinomas vs. ZsGreen-sorted beta cells (ZS beta)[25], Newport green-sorted beta cells (NG beta)[24] and beta cells from insulin-antibody sorting (IS beta)[23]. Note that insulinomas generally differ from normal beta cells regardless of method of beta cell sorting. **b** Differentially expressed protein-coding genes in insulinomas as compared to beta cells. GCG and PPY refer to glucagon and pancreatic polypeptide Y mRNAs, respectively, and reflect the contamination of FACS-sorted beta cells with alpha- and PP-cells. Ins refers to insulin mRNA, is expressed equally in beta cells and insulinomas. **c** The top 20 genes containing the most significant differentially spliced exons, with each column referring to a single sample and color matching the sample color schema in **a**. Expression levels are shown in the color key, with *orange* indicating an increase in insulinomas vs. beta cells. Regardless of beta cell sorting method used, abnormal exon splicing in insulinomas is common. **d** The ten key modules in the insulinoma WGCNA via projection of DEGs. The complete network is shown in Supplementary Fig. 2. **e** Annotation of the ten key co-expression modules, including statistics for DEG enrichment and for the top enriched GO and KEGG terms (at FDR < 0.05)

review of all the variants, a total of 258 somatic (tumor-specific) single-nucleotide variants (SNVs) and 20 non-SNV variants, including indels and multiple-nucleotide variants (MNVs), that alter protein sequence were identified (Supplementary Data 2 and 3), revealing an average of 10.7 such somatic variants per insulinoma exome.

Notably, recurrent variants were rare in insulinomas: only four of 26 insulinomas harbored the previously reported T372R SNV in the *YY1* gene[8, 11, 12]. Only two tumors had *MEN1* mutations, one somatic and one germline. We also identified several novel recurrently mutated genes in insulinomas, including H3 histone family 3A (*H3F3A*; two tumors), lysine-specific demethylase 6A (*KDM6A*; two tumors), Filamin C gamma (*FLNC*; two tumors), ATR serine/threonine kinase (*ATR*; two tumors), and Rho GTPase-activating protein 35 (*ARHGAP35*; two tumors). Interestingly, the seven recurrently mutated genes identified in 10 of

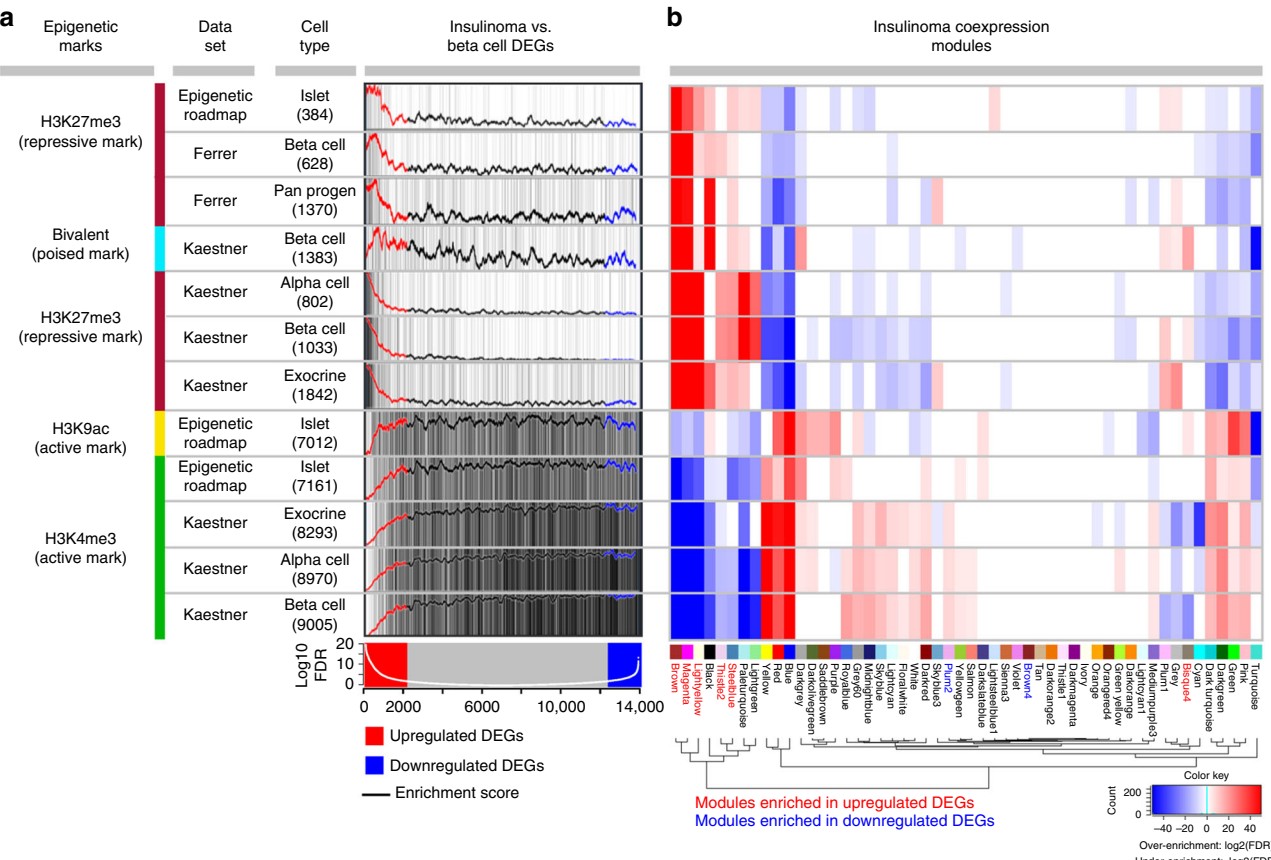

**Fig. 3** Global epigenetic dysregulation in insulinomas revealed by comparison of ~14 K genes from insulinomas and beta cells to histone marks in normal pancreatic cells. **a** H3K4me3, H3K9ac (active) and H3K27me3 (repressive) marks in normal beta cells and other islet-cell types from three independent data sources[27–29] in the top 12 panels are compared to DEGs (insulinomas vs. beta cells) in the *bottom panel*. In the *top panel*, the numbers in the "Cell Type" column represent the number of genes included in each histone mark signature that were also tested in the differential expression analysis from the *bottom panel*, and the *black vertical lines* in the "Insulinoma vs. Beta Cell DEG" column indicate genes with histone marks. In the *bottom panel*, the *red box* indicates genes upregulated in insulinomas and *blue* represents genes downregulated. The *white line* represents –log10(FDR) of the DE analysis. As can be seen, upregulated genes in insulinomas (*red*), in contrast to downregulated DEGs (*blue*) and non-DEGs (*gray*), were highly significantly over-enriched for the repressive mark, H3K27me3, and under-enriched for the active marks, H3K4me3, H3K9ac across the three epigenetic data sources. The histone mark signature visualization curves in the top 12 panels represent the average scores for each sliding window, as described in Methods. **b** Projection of all 12 histone signatures onto the insulinoma co-expression network reveals that the upregulated DEG-enriched modules (*red labels*) were also significantly over-enriched for the repressive mark, H3K27me3, and under-enriched for the active marks, H3K4me3, H3K9ac, at FDR of 0.05, while the downregulated DEG enriched modules (*blue labels*) were generally not enriched for histone signatures. Significant enrichment (FDR < 0.05) is shown in *red* for over-enrichment or *blue* for under-enrichment

the 26 insulinomas (five being epigenetic regulators: *YY1, MEN1, H3F3A, KDM6A, ATR*) are significantly enriched for epigenetic regulators as defined by the curated database, EpiFactors[13] (fold enrichment = 16.7, $p = 2.8 \times 10^{-6}$; Fisher's exact test). Key driver analysis[14, 15] nominated 92 genes, including the seven recurrently mutated genes noted above, as potential disease-causing key driver mutations (Fig. 1a–c, Supplementary Table 2) and revealed similar enrichment for epigenetic regulators (fold enrichment = 2.8, $p = 0.002$; Fisher's exact test). Furthermore, gene ontology (GO) molecular function pathway enrichment analysis of the predicted key drivers revealed similar biological processes of "chromatin-binding" (AdjP = 0.005) among the most significantly enriched terms along with and "SMAD-binding" (AdjP = $1.6 \times 10^{-6}$) (Fig. 1d, Supplementary Data 4). Thus, surprisingly, despite having largely excluded *MEN1*-associated insulinomas, the strongest recurrent mutational signal nonetheless arose from genes encoding epigenetic modifiers functionally related to *MEN1*, which also encodes an epigenetic modifying enzyme. Interestingly, "developmental pathways" were also prominent in GO biological processes, likely reflecting the major role of

chromatin-modifying enzymes in development and differentiation.

Somatic copy number variants (CNVs) including gain, loss, or copy-neutral loss of heterozygosity (cnLOH) were also investigated. A Circos plot summarizing the recurrence of CNVs, identified by the saasCNV algorithm[16], across all 26 insulinomas revealed that chromosome 7 and 11 had the most frequent gain-of-copy and loss-of-copy/cnLOH CNVs, respectively (Fig. 1e). GISTIC2.0 analysis[17] confirmed that some of the strongest amplification and deletion signals arose from chromosomes 7 and 11, respectively, and further defined regions of significant focal CNV events throughout the genome (Fig. 1f, Supplementary Data 5 and 6). Interestingly, *MEN1* resides in one of the significantly deleted regions of chromosome 11q13.1, in 8 of 26 insulinomas via loss-of-copy/cnLOH events (Fig. 1b, f), consistent with previous *MEN1* studies[7, 18–20]. In addition, frequent loss-of-copy/cnLOH of *CDKN1C*, which resides on chromosome 11p15 and encodes the cell cycle inhibitor p57[KIP2], was also observed in 9 of 26 insulinomas (Fig. 1b), as reported previously in pediatric insulinomas[18]. Pursuing the "epigenetic modifier" gene ontology

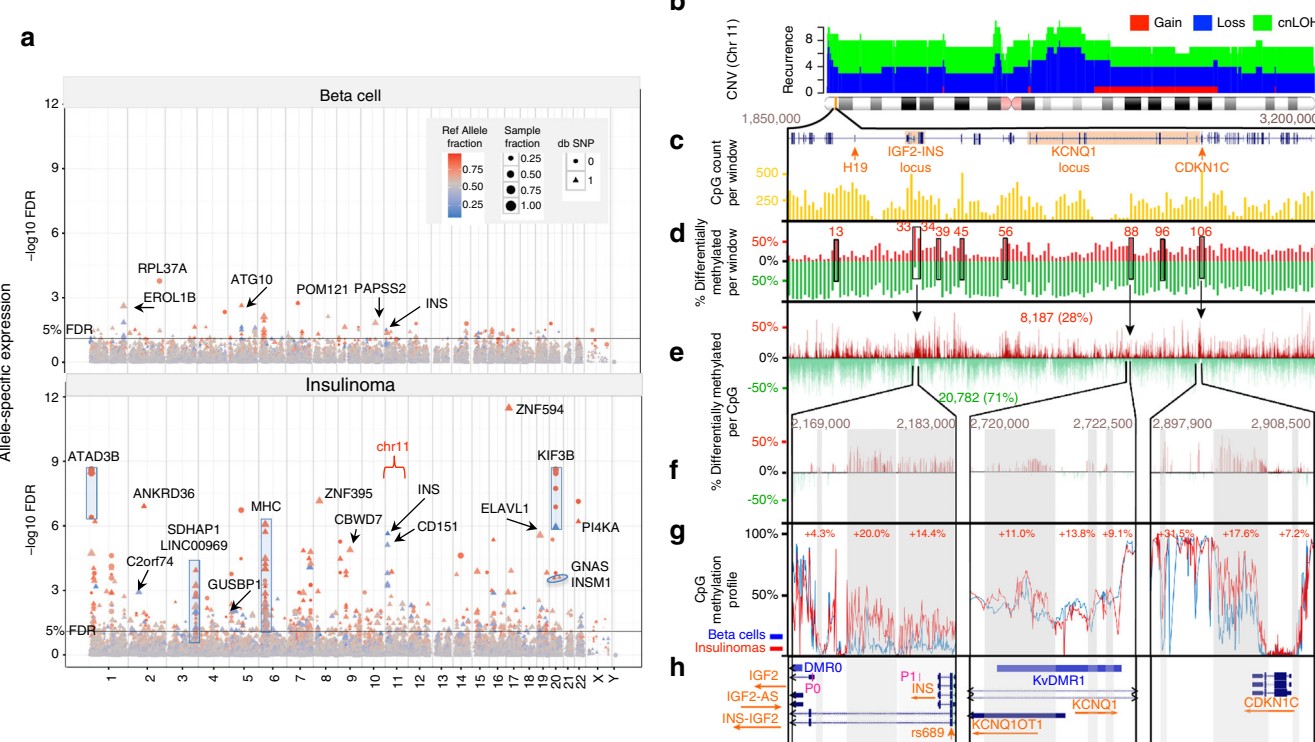

**Fig. 4** Global allele-specific expression and CpG methylation patterns in insulinomas and beta cells suggests a link between CNVs and ASE on the chromosome 11 imprinted locus. **a** A Manhattan plot of ASE shows readily apparent genome-wide differences between beta cells (*top panel*) and insulinomas (*lower panel*), suggesting broad genomic ASE abnormalities in insulinomas. Most of the differences observed are in the reference allele (*red*). However, in insulinomas there is a strong preference for ASE from the alternate allele (*blue*) in chromosome 11 near the insulin (INS) locus. **b** CNV analysis suggests that chromosome 11 harbors recurrent loss/cnLOH with each CNV segment recurring in an average of 8–9 of the 26 insulinomas. In addition, one of the most significant deleted regions identified by GISTIC2.0 analysis (Fig. 1f) is found on cytoband 11p15.5, which also contains the two imprinting control regions (ICRs). **c** A blow up of the 11p15.5-15.4 imprinted region that underwent DNA methylation analysis, highlighting key 11p15 genes and showing the density of the 29,675 CpGs in the 1.35 Mbp, in 135 windows of 10Kbp. CpG density in the region in bins of 10 kb. **d** A comparison of methylation patterns between insulinomas (n = 10) and FACS-sorted beta cells (n = 2), in the same 10 kb windows as in **c**. Insulinomas display at least nine hypermethylated windows (*red*) denoted W13-106. Note also the broad hypomethylation (*green*) in this region. In all, 70.4% of CpGs were hypomethylated and 27.7% hypermethylated, with 1.9% showing no differential methylation. **e** Expanded view showing the relative methylation of the 29,675 CpG in the 11p15 region, highlighting four windows (W33, 34, 88,106) with the greatest relative hypermethylation in insulinomas. **f** Further expansion of the three hypermethylated regions in **e** delineating individual hypermethylated CpGs. The *gray shaded areas* highlight the hypermethylated regions. **g** The average CpG methylation status of each of the CpGs in the three regions in insulinomas (*red*) vs. beta cells (*blue*). **h** Genome browser view of the three highlighted 11p15 regions, highlighting the genes and regions affected by the hypermethylated CpGs in the insulinomas, including *IGF2, INS-IGF2, IGF2-AS, KCNQ1, KCNQ1OT*, the imprinting control region KvDMR, and *CDKN1C*

signature from the earlier mutational analysis, we also surveyed CNVs harboring other epigenetic modifiers in addition to *MEN1*. Among this class, the *EZH2* locus on chromosome 7 (7q36.1) displayed a gain of copy CNV in 10 of 26 insulinomas (Fig. 1b). Thus, collectively, at the DNA level, 22 of 26 insulinomas (85%) contain recurrent SNVs/MNVs/indels (*YY1, H3F3A, KDM6A, MEN1, ATR*) and/or recurrent CNVs (*MEN1, EZH2*) in genes encoding epigenetic modifiers (Fig. 1c), four of which are members of two key epigenetic modifying complexes, the polycomb group (*EZH2, YY1*) and the trithorax group (*MEN1, KDM6A*). These complexes are recruited to specific regions of DNA and direct the post-translational modification of histones (e.g., H3F3A) to either repress (polycomb) or activate (trithorax) gene expression[21, 22].

**Transcriptomic networks reveal pathways underlying growth.** RNAseq analysis was performed on 25 insulinoma samples, 13 of which had matching WES analysis (Supplementary Data 1 and 7). We compared the transcriptomes of the insulinomas to those of beta cells isolated using three independent methodologies from 22

normal human cadaveric islet donors[23–25] (Supplementary Data 7). Unsupervised classification analysis, after adjustment for gender and library platform covariates, showed that insulinomas cluster separately from beta cells, regardless of isolation technique (Fig. 2a).

Comparison of insulinoma to beta cell transcriptomes revealed 3709 differentially expressed protein-coding genes (DEGs) (FDR < 0.01) (Fig. 2b, Supplementary Data 8) of which 2125 were upregulated and 1584 were downregulated in insulinomas. Notably, insulin was consistently the most highly expressed transcript across all samples, confirming the identity of the PNETs as insulinomas. We further inferred differential splicing in insulinomas vs. beta cells through analysis of differential exon usage/retention (Fig. 2c, Supplementary Data 9 and 10). For example, exon-level analysis identified the major histocompatibility II transcriptional regulator, *CIITA* (Simes Test FDR = $4.2 \times 10^{-27}$), which was also upregulated at the DEG level in insulinomas, as being among the top 10 most differentially spliced genes (DSGs) in insulinomas (Supplementary Data 10).

As part of a data dimensionality reduction approach to identify the key modules of genes that underlie the altered biological

processes occurring in insulinomas relative to beta cells, a signed, weighted gene co-expression network analysis (WGCNA)[26] was performed (Supplementary Data 11, Supplementary Fig. 2). Projection of the DEGs (insulinoma vs. beta cells) onto the insulinoma-only co-expression network (52 co-expressed modules), yielded six modules that were over-enriched for upregulated genes, and two modules were significantly over-enriched for downregulated genes (cutoff: fold-enrichment >2, FDR < 0.01) (Fig. 2d, e, Supplementary Data 12). Two modules (red and grey60) were found to be significantly under-enriched for both upregulated and downregulated genes (fold enrichment <0.6, FDR < 0.01) (Fig. 2d, e). Pathway enrichment analysis based on GO and KEGG databases was used to annotate the biological functions of each insulinoma co-expression module (Fig. 2e, Supplementary Data 13). Key biologies highlighted by the DEG-enriched modules included "immune system", "extracellular matrix", "vasculature development", "cell proliferation", "RNA splicing" and "ubiquitination". This diverse biology indicates a complex transformation of the insulinoma, despite the conserved function of insulin secretion.

**Insulinoma transcriptomes link to epigenetic dysregulation**. Since the genomic evidence indicated a unifying theme of perturbation of epigenetic modifiers in insulinoma, including modifiers of histone methylation (Fig. 1), we looked for hints of altered chromatin states regulating the insulinoma transcriptome. We focused on histone modifications, in particular histone-H3 post-translational modifications, by testing the enrichment of DEGs (insulinoma vs. beta cell) across prior reports in which three histone mark signatures (H3K27me3, H3K4me3, H3K9Ac) had been characterized in normal human beta cells or other islet cells[27–29] (Fig. 3a, Supplementary Data 14). Remarkably, the upregulated DEGs in insulinomas were significantly over-enriched for the repressive mark (H3K27me3) in normal beta cells (Ferrer beta[29]: fold enrichment = 3.3 and FDR $3.1 \times 10^{-37}$; and Kaestner beta[28]: fold enrichment = 9.0, FDR = $2.6 \times 10^{-216}$) and under-enriched for the active mark (H3K4me3) (Kaestner[28] beta: fold enrichment = 0.2, FDR = $4.7 \times 10^{-222}$). This striking contrast between genes upregulated in insulinomas vs. normal beta cells bearing the repressive mark was consistent across the three different data sources. This trend also extended to non-beta cells: alpha cells, exocrine cells and pancreatic progenitor cells, as well as whole islets. The reverse scenario (under-enrichment for the active histone mark, H3K4me3, in beta cells) was also evident for upregulated DEGs in insulinomas (Fig. 3a, Supplementary Data 14). Furthermore, the histone mark signature visualization scores revealed that even among the upregulated DEGs, the DEGs with more significant FDRs (the *white line* in the *bottom track* in Fig. 3a) were more likely to have a high score for H3K27me3-signatures and low score for H3K4me3 signatures (Fig. 3a). To summarize, precisely the same collection of ~ 2000 genes that are repressed in normal beta cells (and related pancreatic and endocrine cells) and which bear strong H3K27me3 repressive chromatin marks and weak H3K4me3 open chromatin marks in beta cells, are overexpressed in insulinomas; this is true whether genes are assessed individually (Fig. 3a) or as modules of co-expressed genes (Fig. 3b). These findings provide compelling evidence to support a functional role for the recurrent mutations in polycomb and trithorax genes in the broadly abnormal gene expression in insulinomas.

To explore further the different beta cell histone mark signatures in the context of insulinoma transcriptome, we projected the same histone mark signatures onto the insulinoma co-expression network. Importantly, the same modules over-enriched for upregulated DEGs were also significantly over-enriched for the repressive mark (H3K27me3) and under-enriched for the active marks (H3K4me3, H3K9ac) (FDR < 0.05). The downregulated DEG-enriched modules, in contrast, were generally not enriched for any specific histone signatures (Fig. 3b, Supplementary Data 15). Thus, a significant set of genes that are upregulated in insulinomas relative to beta cells appear to be de-repressed through epigenetic modifications that are recorded by trithorax and polycomb family members.

**Global abnormalities in allele-specific expression**. The epigenetic aberrations described above may have additional consequences, among which may be changes in allele-specific expression (ASE). Lacking parental genotype information, we interrogated the insulinoma and beta cell RNAseq data to identify genes subject to ASE. Across the entire insulinoma genome, we detected 717 loci (586 of which were common SNPs) showing potential allelic imbalance (FDR < 5%) (Supplementary Data 16). A Manhattan plot of these data (Fig. 4a) reveals a strong allelic bias in the insulinomas relative to beta cells, largely involving loci not previously reported in beta cells[30].

Although the present data set cannot dissociate these ASE results from epigenetic vs. genetic (e.g., cis-eQTL) causation, they nonetheless provide additional insights into insulinoma biologies. Strikingly, ASE highlighted the well-known imprinted region of chromosome 11 as preferring mono-allelic expression (Fig. 4a). For example, 8 of 24 insulinomas revealed a significant (FDR < 0.01%) preference for the alternate "T" allele at the SNP, rs689. This SNP is located at the 5' end of the *INS-IGF2* locus, near the differentially methylated regions, DMR0 and DMR1. These DMRs assist the imprinting control region (ICR) located immediately upstream of *H19* to enforce ASE, by the parent-of-origin, of maternally expressed *H19* and paternally expressed *IGF2* in chromosomal region 11p15.5. The rs689 allelic imbalance was also observed in 5 of 22 beta cell samples, albeit at much lower significance (FDR < 3%), consistent with a previous report[24]. To further understand the possible impact of the rs689 SNP in the pathobiology of human insulinoma, we explored the relationship between the A and T alleles and expression of transcripts in this region. There was no impact on expression of *H19, IGF2, INS*, but usage of the alternate T allele was associated with significantly higher expression of the *INS-IGF2* anti-sense transcript, and a trend toward higher expression of the *INS-IGF2* read-through transcript (Supplementary Fig. 3, Supplementary Data 17). Analysis of RNAseq profiles in insulinomas expressing the T vs. A alleles at rs689 revealed multiple DEGs, the gene ontology analysis of which suggested that usage of the T allele favors a neural signature in insulinomas (Supplementary Fig. 3).

**Altered DNA methylation patterns at 11p15 in insulinomas**. Because 11p15 is a paradigmatic imprinted region, because 11p15 imprinting abnormalities are associated with beta cell proliferation in the focal variant of congenital hyperinsulinism (FoCHI) and the Beckwith–Wiedemann Syndrome (BWS)[31–34], and because insulinomas display marked reductions in the cell cycle inhibitor, p57[KIP2] encoded by *CDKN1C* in the 11p15 imprinted region (see below), we performed deep CpG methylome sequencing in the 11p15.5-p15.4 region on 10 insulinomas and compared this to the methylation pattern in two sets of FACS-sorted normal human beta cell preparations (Fig. 4b–h). As compared to normal beta cells, insulinomas display broad and widespread hypomethylation, together with a hypermethylation in a few selected areas. These regions contain sequences known to have important regulatory functions in imprinting regulation, possibly affecting ASE, and which may hint at underlying disordered 3-D chromatin structure in this region. Insulinomas also

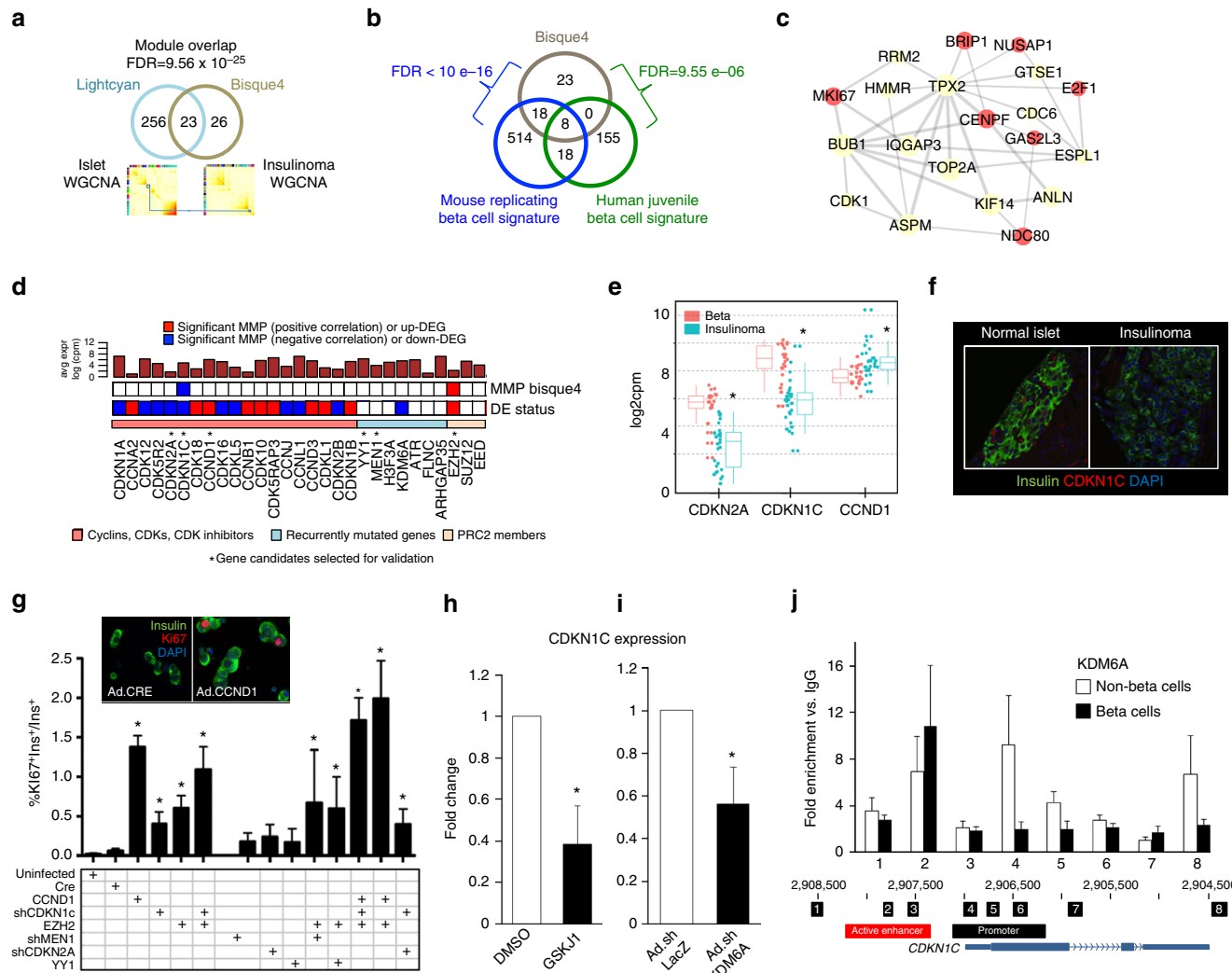

**Fig. 5** Comparison of the bisque4 insulinoma proliferation module with adult human islets, replicating normal human juvenile and mouse beta cells, validation of gene candidates for beta cell proliferation, and direct modulation of *CDKN1C* by trithorax member, KDM6A. **a** The insulinoma proliferative bisque4 module is preserved in the islet co-expression network. **b** The bisque4 module is the only significantly over-enriched module (at FDR = 0.01) via projection of gene signatures of human juvenile beta cells and from replicating mouse beta cells onto insulinoma co-expression network. **c** The top 20 "hub genes" in the bisque4 module based on intra-modular connectivity. *Red balls* indicate differentially upregulated genes (FDR < 0.01); *yellow balls*, non-significant. Increased line weights indicate stronger connectivity. **d** A panel of predicted candidate genes as possible mediators of beta cell proliferation. The top bar graph shows their level of average expression based on RNAseq (log2CPM). The *first row* shows their bisque4 MMP status; the *second row* shows their DE status in insulinomas vs. beta cells. Genes selected for validation are marked with an *asterisk*. **e** Differential expression from RNAseq for *CDKN2A, CDKN1C*, and *CCND1* in insulinomas vs. beta cells. **f** A representative immunohistochemical photomicrograph showing p57$^{KIP2}$ (*red*) is readily detectable in approximately half of beta cell (*green*) nuclei, but undetectable in non-beta cell islet cell types and insulinomas. **g** Induction of proliferation in human beta cells by adenoviral overexpression or silencing of the genes shown. Each *bar* represents the mean of 4–10 different human islet preparations. *Error bars* indicate mean ± SEM; *asterisks* indicate *p* < 0.05 by Student's *t*-test. Note that expression of *CCND1* or *EZH2*, or silencing of *CDKN1C* individually all induce human beta cells to enter cell cycle at rates comparable to those observed in insulinomas, and that *EZH2-shCDKN1C* combination appears additive. In contrast, silencing *MEN1*, or overexpressing wild-type or mutant *YY1* fails to activate proliferation under the conditions studied. *Asterisks* indicate *p* < 0.05 by Student's *t*-test. The *inset* shows examples of Ki67 immunolabeling (*red*) in human beta cells (*green*) induced by a control adenovirus encoding Cre recombinase and adenoviruses encoding *CCND1*. Similar effects were seen with EZH2 or shCDKN1C (not shown). **h** Repression of *CDKN1C* expression in human islets by the KDM6A inhibitor, GSKJ1. *Error bars* indicate mean ± SEM of five human islet preps; *asterisks* indicate *p* < 0.05 by Student's *t*-test. **i** Repression of *CDKN1C* expression by adenoviral silencing of *KDM6A* in human islets. *Error bars* indicate mean ± SEM of five human islet preps; *asterisks* indicate *p* < 0.05 by Student's *t*-test. **j** Top panel: ChIP analysis of KDM6A binding to the *CDKN1C* locus in five pairs of FACS-sorted human beta cells and non-beta cells. *Error bars* indicate mean ± SEM. Bottom panel: A genome browser view of the *CDKN1C* locus highlighting primer pairs 1–8[43], promoters, enhancers, exons, and transcriptional start site of *CDKN1C*. In beta cells, KDM6A binds to *CDKN1C*, predominantly at an upstream enhancer delineated by primer pair #2. KDM6A also binds to *CDKN1C* in non-beta cells of the islet. Note that the directionality of this panel is opposite that in Fig. 4h

display CNV loss, copy-neutral LOH in the 11p15 region over-lapping with key regulatory elements (e.g., transcriptional start sites, enhancers, CTCF-binding sites, ICRs, splice sites) of important transcripts in this region including *INS, INS-IGF2, IGF2, KCNQ1, KCNQ1OT, H19,* and *CDKN1C.* This is corroborated by marked overexpression of *H19* and *KCNQ1* and underexpression of *CDKN1C* in the large majority of insulinomas, as described in more detail below. IGF2 was upregulated in several insulinomas, but mostly unchanged compared to beta cells (Supplementary Data 8). Notably, despite the widespread CpG abnormalities in this region, *INS* expression is preserved in insulinomas. The adjacent 11p15.4 region also contains the non-imprinted genes, *ABCC8* and *KCNJ11*, which encode the sulfonylurea receptor, SUR1, and the Kir6.2 potassium inward rectifier, respectively. These were not significantly differentially expressed (Supplementary Data 8). Collectively, the widespread and recurrent structural and/or imprinting abnormalities within 11p15—already strongly associated with pathologic beta cell proliferation in FoCHI and the BWS syndrome[31–34]—suggest an important role for abnormalities in this region in the molecular pathogenesis of insulinoma. This should be more deeply explored in the future using 3-D structural approaches such as CTCF ChIPseq, 3-C and 5-C chromatin capture studies.

**An integrative approach suggests proliferation drivers**. Our integrative genomics approach led to the identification of ten co-expression modules (Fig. 2e) that may underlie insulinoma pathogenesis, many of which were linked to potential epigenetic dysregulation (Fig. 3b). Given our interest in beta cell regeneration, and seeking the mechanisms that may permit escape from beta cell quiescence in benign insulinomas, we elected to focus on the bisque4 co-expression module, the module in insulinomas that contained DEGs enriched for cell proliferation. Furthermore, the bisque4 cell cycle module was particularly enriched for beta-cell specific histone mark signature, including genes with bivalent marks, both H3K27me3 and H3K4me3 (Kaestner beta[28]: fold-enrichment = 5.2, FDR = $4.0 \times 10^{-6}$) (Fig. 3a,b).

To assess the relevance of the bisque4 module to normal beta cell proliferation, we compared the co-expression of the genes in the bisque4 module to those within normal beta cells. Similarly as for insulinomas, WGCNA was performed for human islets from a large publicly available data set of 89 samples[30], the majority of which were from healthy adults (Supplementary Fig. 4). We reasoned that an islet-based co-expression network could serve as a proxy for a normal beta cell co-expression network, given beta cells on average make up approximately 50% of the human islet cell population[35, 36]. Importantly, the insulinoma bisque4 cell cycle module was found to significantly overlap with that of only one islet co-expression module, the lightcyan module (fold enrichment = 23.7, FDR = $9.6 \times 10^{-25}$, Fig. 5a). This suggests that co-expression of cell cycle genes in normal beta cells is preserved in insulinomas, despite their greater rate of proliferation.

Seeking to model the normal physiologic beta cell expansion seen in childhood, we further projected two independent gene expression signatures derived from proliferating juvenile human[37] and mouse[38] beta cells onto the insulinoma co-expression network. Among these many modules, only the bisque4 module was over-enriched (human[37]: fold enrichment = 12.7, FDR = $9.6 \times 10^{-6}$; mouse[38]: fold enrichment = 13.4, FDR < $1.0 \times 10^{-16}$) (Fig. 5b). Altogether, these observations suggest a commonality in the mechanisms that underlie proliferation in insulinomas compared to those in juvenile mouse and human beta cells.

Importantly, among the top 20 hub genes (based on intra-modular connectivity)[26] in the bisque4 module were genes

directly involved in beta cell cycle progression (Fig. 5c), including *E2F1, MKi67* and *CDK1.* However, since many of the bisque4 module genes were found to be related to late ($G_2$/M) cell cycle stages (Supplementary Data 11), we searched for genes farther upstream of the $G_0$/$G_1$ transition by expanding the bisque4 module to include genes whose expression in insulinomas was significantly correlated with the bisque4 module gene expression profiles, thereby assigning a "bisque4 module membership $p$-value" (MMP). At a bisque4 MMP of 0.01, an additional ∼ 200 genes were revealed (Supplementary Data 18).

We first surveyed differentially expressed (insulinoma vs. beta cells) cyclins, cyclin-dependent kinases (CDKs) and CDK inhibitors for their MMP status (Fig. 5d). Interestingly, *CDKN1C*/p57[KIP2] was the only CDK inhibitor with significant bisque4 MMP, nominating it as a key candidate driver of the proliferative phenotype in human insulinomas. The consistent reduction of *CDKN1C* relative to normal beta cells was confirmed both by DEG analysis as well as immunohistochemistry on a human insulinoma tissue microarray (Fig. 5e, f), and supports the putative imprinting abnormalities and CNV loss in the *CDKN1C* imprinted region on chromosome 11p15.5 (Fig. 4b). Moreover, a central role for *CDKN1C*/p57[KIP2] is supported by its well documented loss in FoCHI, BWS, and pediatric insulinomas syndromes[18, 31–34], and by the observation that lentiviral silencing of *CDKN1C*/p57[KIP2] in transplanted human islets leads to proliferation[39]. Notably, among differentially expressed genes encoding G1/S phase cell cycle activators, two of the most highly expressed were *CCND1* and *CCND3* (Fig. 5d, e). We have reported previously that *CCND1* is increased in 40% of insulinomas[40], and that overexpression of either *CCND1* or *CCND3* leads to human beta cell proliferation[41].

In addition, we explored the recurrently mutated genes, genes encoding polycomb and trithorax complex members for their bisque4 MMP and DEG status (Fig. 5d). Interestingly, among the polycomb genes, only *EZH2* achieved a significant bisque4 MMP (Spearman correlation = 0.60, $p = 0.002$), concordant with the observation that in mouse models, *Ezh2* epigenetically represses *CDKN2A*/p16[INK4A] in beta cells, and is requisite for beta cell proliferation in juvenile mice[42]. As noted earlier, *EZH2* upregulation in some insulinomas may arise from the increase in copy number at the *EZH2* locus on chromosome 7q. The observations that *CDKN1C* (rather than *CDKN2A*) was the CDK inhibitor best co-correlated with bisque4, and that *EZH2* can transcriptionally repress *CDKN1C*[43] suggest that combined *EZH2* and *CDKN1C* loss/repression may be critical to enable proliferation in insulinoma cells.

**Biological validation of candidate proliferation drivers**. To validate combinations of lead candidate genes derived from our integrative approach as human beta cell mitogenic mediators, from the list of ∼ 200 bisque4-associated genes, we developed adenoviruses to overexpress *EZH2* and *YY1* (mutant and wild-type), to silence *MEN1* or *CDKN1C*, alone or in various combinations, and to determine which, if any, candidates might induce replication in human beta cells (Fig. 5g). Surprisingly, neither silencing *MEN1* nor overexpressing mutant or wild-type *YY1*-induced proliferation, perhaps reflecting a requirement for additional mitogenic events, or longer lead time to induce requisite epigenetic changes. In contrast, adenoviral over-expression of *CCND1* or *EZH2*, or silencing of *CDKN1C*, alone and in several combinations, were all capable of inducing human beta cell replication, at rates mimicking the 1–2% Ki67 labeling index characteristic of benign human insulinomas[44]. Importantly, overexpression of *EZH2* in combination with silencing *CDKN1C* was more effective than either experimental perturbation alone.

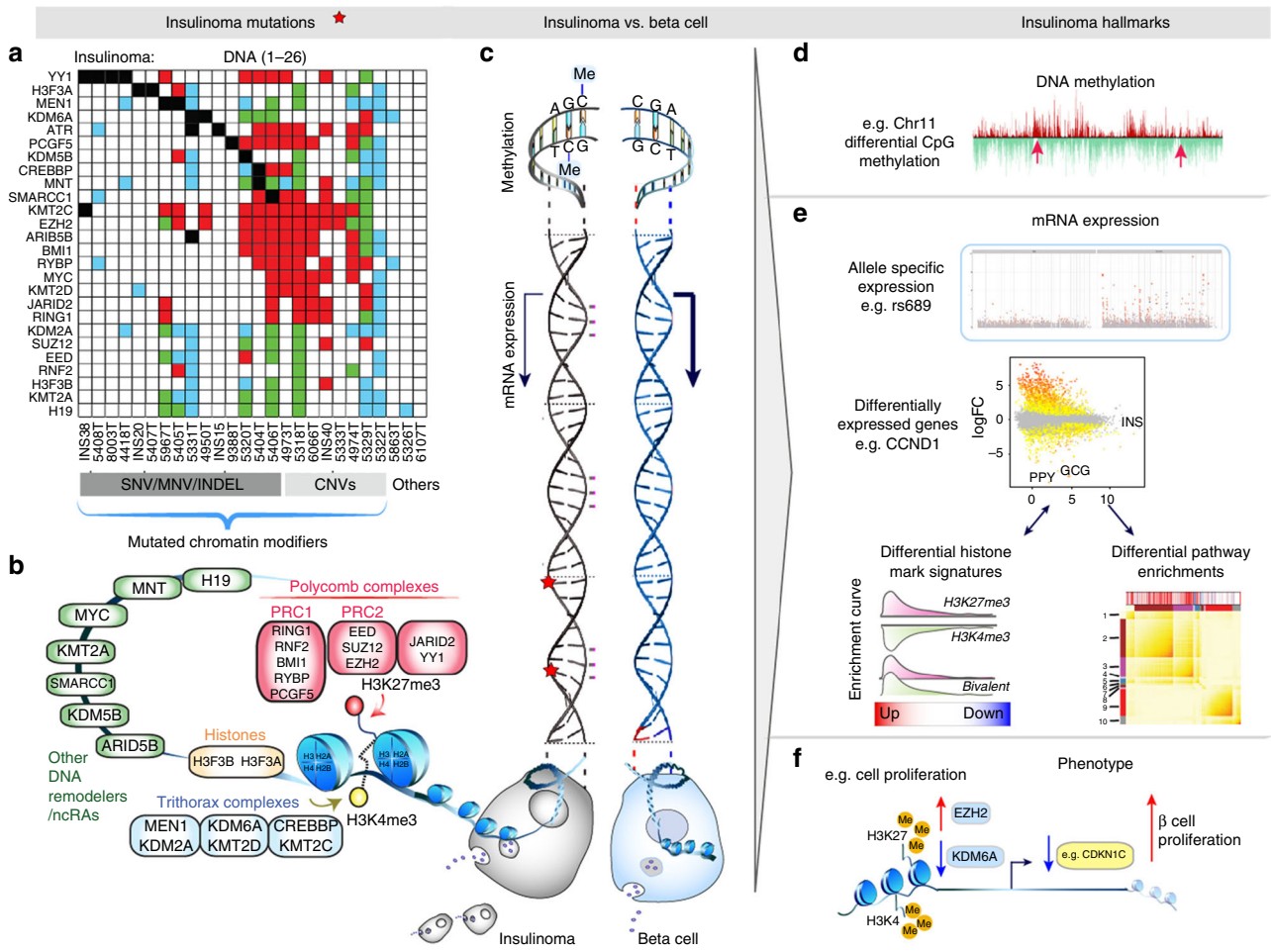

**Fig. 6** A schema summarizing the broad heterogeneity of insulinomas and the molecular hallmarks that distinguish insulinoma from beta cells. **a** A waterfall plot summarizing cumulative coding mis-sense variants (*black*), CNV loss (*blue*), copy-neutral LOH (*green*), or copy number gain (*red*) among polycomb, trithorax and related chromatin-modifying genes. In the *grid boxes*, each *vertical column* represents the DNA landscape of each of the 26 insulinomas. Collectively, these observations highlight the frequency of multiple genomic and transcriptomic abnormalities in polycomb, trithorax, and other epigenetic modifying genes across the majority of human insulinomas, the marked heterogeneity among insulinomas, the frequent mutation of multiple chromatin modifiers in almost all insulinomas, all despite having selected insulinomas independent of known MEN1 kindreds. **b** A cartoon emphasizing the relationship between polycomb, trithorax genes, and H3K27me3 and H3K4me3 marks in human beta cells and insulinomas. **c** An extension of **b** highlighting mutations (*red stars*), and abnormal CpG methylation. Subsequent panels highlight altered DNA methylation/imprinting abnormalities (**d**), asymmetrical gene expression from imprinted loci and abnormal chromatin marking patterns (**e**), all of which may lead to differential expression pattern of cell cycle genes such as *CDKN1C* (**f**), all of which may lead to increased beta cell proliferation in insulinomas

The *EZH2-CCND1* combination yielded the highest Ki67 labeling indices of all. As expected, overexpression of *CCND1* also induced Ki67 labeling in alpha, delta and PP cells in dispersed islets (Supplementary Fig. 5). The other combinations had no effect on delta or PP cells, whereas EZH2 overexpression in combination with silencing *CDKN1C* did low level induce Ki67 labeling in alpha cells.

**Trithorax dysfunction leads directly to *CDKN1C* reduction**. To determine if a direct link exists between trithorax members and cell cycle machinery in human beta cells, we selected the H3K27me3 demethylase, *KDM6A*, for further study, since it is a canonical trithorax member, since it is one of the rare recurrently mutated genes in insulinoma (Fig. 1), since it's expression is reduced in insulinomas, (Supplementary Data 8) and since its biology has not previously been explored in human beta cells. We selected *CDKN1C* as a target because it is the only cell cycle inhibitor in the bisque4 MMP list (Fig. 5d), because it is reproducibly reduced in insulinomas (Fig. 5e, f), because its loss is

associated with beta cell proliferation in FoCHI and BWS[31–34], because *CDKN1C* is expressed only in beta cells in human islets[23–25, 32], and, because silencing *CDKN1C* leads to induction of human beta cell proliferation in our hands (Fig. 5g) and others[39]. Inhibition of *KDM6A*, whether pharmacologically with GSKJ1 or adenovirally with sh*KDM6A*, led to prompt and substantial reduction of *CDKN1C* in human islets, an effect not observed by adenoviral silencing the closely related *KDM5B* or *KDM6B* (Fig. 5h, i, Supplementary Fig. 6a–d). Chromatin immunoprecipitation (ChIP) analysis demonstrated that KDM6A directly interacts with a *CDKN1C* upstream enhancer in human beta cells (Fig. 5j). Further, the *CDKN1C* locus in human beta cells bears the H3K4me3 open chromatin mark, but lacks the H3K27me3 repressive chromatin mark, as anticipated by trithorax occupancy (Supplementary Fig. 6e). Collectively, these results reveal that KDM6A, and by extension, the trithorax complex, directly support *CDKN1C* expression in human beta cells, and, conversely, that interfering with trithorax function results in *CDKN1C* loss, with consequent cell cycle entry. Notably, they also suggest yet another pharmacological approach to

induction of human beta cell proliferation (Supplementary Fig. 6f).

## Discussion

Overall, this integrated and systematic analysis of a large human insulinoma cohort, focused on non-MEN1 insulinomas, reveals a near-universal preponderance of mutations in, and differential expression of, epigenetic modifying genes and their targets, together with structural and CpG methylation alterations of the key imprinted region on chromosome 11 (Fig. 6). Genomic mutations were particularly notable in genes related to the polycomb and trithorax complexes. While inactivation or loss of the trithorax complex member, menin, encoded by *MEN1*, is a well-known cause of insulinoma in mice and humans, the striking prevalence of mutations and/or dysregulation in many other trithorax and polycomb members in non-MEN1 insulinomas was not anticipated. In retrospect, it aligns nicely with prior work[37, 38, 45–47], which suggest that the epigenome, through its broad control of cell cycle and function-modulating beta cell genes, represents the key rate limiting step in human beta cell proliferation.

Within this unifying context, the model also illustrates that multiple specific "epigenetic roads" may lead to insulinoma, and conversely, that despite their apparent clinical homogeneity, insulinomas display marked mutational heterogeneity, which, in retrospect, could only have been observed in a large number of insulinomas such as described herein. The simplest of these heterogeneous paths might be loss of the maternal chromosome 11, or chromosome 11 uniparental paternal disomy, as occurs in some cases of FoCHI, BWS, and pediatric insulinoma syndromes[18, 31–34]. Chromosome 11 errors, suggested by recurrent loss of this region and the unique ASE profile, as well as imprinting errors directly confirmed by deep CpG methylome sequencing, with congruent increases in *H19, KCNQ1*, and recurrent loss of *CDKN1C* (Fig. 4), also likely contribute to a large subset of insulinomas. Interestingly, the insulinomas with chromosome 11 loss are distinct from those with chromosome 7 gain, suggesting that these may represent two insulinoma subtypes. In the latter, the increase in *EZH2* relative to normal beta cells may result from a CNV gain event. Importantly, the average of 3.5 driver mutations and multiple additional CNV gains and losses that characterize insulinomas (Figs. 1 and 6) is congruent with the observation that *MEN1*-associated human insulinomas occur only after decades[5–7], and that loss of both *men1* alleles in mice nonetheless requires months of latency before insulinomas appear[48]. Thus, it appears inescapable that mutations in single genes such as *MEN1* alone cannot cause insulinoma; instead, "hits" in multiple genes are likely required: *MEN1* is just one of many core polycomb/trithorax genes (e.g., *MEN1, KDM6A, YY1, EZH2, PCGF5, KTM2C, CREBBP* etc.), which may contribute to the insulinoma phenotype.

The heterogeneity observed among insulinomas is reminiscent of the heterogeneity among human beta cells that has captured recent attention[49–51]. To determine whether insulinomas are also heterogeneous with respect to the hallmark markers of human beta cell heterogeneity, we explored CD9 and ST8SIA1 expression in our beta cell and insulinoma transcriptome data sets (Supplementary Table 10). Indeed, both ST8SIA1 and CD9 expression were highly variable among both human beta cells as well as insulinomas (Supplementary Fig. 7).

Since we are interested in inducing benign, rather than malignant growth of beta cells, it was reassuring to observe parallels between cell cycle genes and modules in insulinomas and proliferating juvenile beta cells (Fig. 5a, b), and also to observe that mutations in DNA repair genes such as *ATRX, DAXX, TP53,*

*BRCA2, CHEK2*, and the PI3 kinase/PTEN/mTor/TSC pathway, characteristic of malignant non-functioning PNETs[52, 53], were not observed in the insulinomas. This may reflect an important role for genes such as *DAXX, ATRX, CHEK2, BRCA2*, and *TP53* in the rare malignant transformation of insulinomas; these genes may also provide clues to genes and pathways that must be avoided in attempts at therapeutic expansion of human beta cells.

Human insulinomas do not retain normal control of glucose sensing and insulin secretion. To explore mechanisms underlying this imperfect control, we surveyed families of so-called "disallowed genes"[54, 55], glucose transporters and hexokinases, and beta cell-specific transcription factors (Supplementary Figs. 8–10). Remarkably, abnormalities were abundant and tantalizing here, including, alterations in glucose transporter and *HK3* profiles, and a suggestion of restitution of expression of certain disallowed genes, although their expression remained at low levels. These observations provide clues to deeper understanding of the misregulation of insulin production in insulinoma. Canonical beta cell transcription factors such as *PDX1, NKX6.1, MAFA*, and others were not substantially altered in insulinomas vs. beta cells. We also explored whether expression of mitogenic genes such as *CCND1, EZH2, YY1*, or silencing *CDKN1C* or *KDM6A* might alter expression of these families of genes in human islets (Supplementary Figs. 8–10). Reassuringly, these maneuvers had little effect, with the exception of overexpression of *YY1*, which altered expression of *PDX1, NKX6.1, SLC16A1*, and *SLC2A2*. These observations may suggest that it is best to avoid *YY1* activation in therapeutic attempts to induce beta cell expansion.

We observed that several insulinoma co-expression modules were enriched not only for cell cycle genes (the bisque4 module), but also enriched for "immune function", "vascularization", "extracellular matrix", "RNA splicing", and "ubiquitination" (Fig. 2d, e). While we chose to focus on the bisque4 cell cycle module, the co-expression analysis highlights the importance of the beta cell microenvironment in supporting insulinoma growth and function. The immune-insulinoma connection was supported by the appearance of *CIITA* among the top DSG in insulinomas (Fig. 2c). Furthermore, many of the *CIITA* transcriptional targets, the MHC class II genes, were also differentially expressed in insulinoma relative to beta cells. Interestingly, YY1 and JARID's are DNA-binding partners, which interact with, and regulate *EZH2* recruitment to, the inducible promoter pIV of *CIITA*[56]. While there have been prior reports linking beta cell proliferation to inflammation in type 1 diabetes[57], this apparent link between *CIITA* and insulinoma is unexplained.

Collectively, this study provides a novel and complex lens through which to view insulinoma and its relationship to normal beta cell function, and raises a number of additional questions noted above that merit further study. For example, what drives the near universal *EZH2* overexpression in insulinomas? The precise genetic and molecular pathophysiology remain unexplained. Similarly, what drives *CCND1* overexpression? Further, these studies do not clearly distinguish between genomic/transcriptomic abnormalities that lead to beta cell proliferation, vs. those that lead to insulin overproduction. Clarification of this issue would be facilitated by a similar comparison of insulinomas to other benign PNETs that do not overproduce insulin. Similarly, it would be informative to compare these non-MEN1 insulinomas to the whole-exomes and transcriptomes of bona fide MEN1-associated insulinomas, to elucidate which features the two different types of tumors share on their passage from normal beta cell to adenoma. Finally, we highlight that previously reported mitogenic targets of the harmine class of beta cell mitogenic drugs[2–4]—*DYRK1A* (from the red module, Fig. 2), and *MYC* (also upregulated in insulinomas, Supplementary Data 8)

and its inhibitor, *MNT* (Fig. 6)—are also present in this analysis. The observation that pharmacologic or adenoviral *KDM6A* interference can downregulate the cell cycle inhibitor, *CDKN1C*, suggest another entirely novel therapeutic target. Altogether, these observations clearly illustrate that "insulinoma data mining" can yield beta cell regenerative targets. Further studies involving whole-genome sequencing of insulinomas, defining lncRNA and microRNA profiles, whole-genome DNA methylation patterns, imprinting marks, histone marks, and chromatin structure, and comparing malignant to non-malignant insulinomas, will deepen the description of the proliferating human beta cell, and provide additional opportunities for therapeutic drug discovery.

## Methods

**Insulinomas**. Thirty-eight human insulinomas were studied. Thirty-four were from subjects who provided informed consent, and were deposited in The Icahn School of Medicine at Mount Sinai Biorepository; WES data from an additional four insulinomas from Cao et al.[8] passed our QC filtering and were included in the sample cohort. Patient samples were de-identified through the Biorepository and Pathology Core at the Icahn School of Medicine and IRB-HSM-00145. Of 26 WES paired whole-blood and tumor insulinoma samples, 22 were sequenced at Mount Sinai, with the other four from Cao et al.[8]. Of the 38 insulinomas, 25 were subject to RNA sequencing; 13 from the same samples that underwent paired WES; 12 did not undergo WES because of a lack of paired WBC samples. All displayed marked increases in insulin RNA expression by RNAseq. Details of the patients with insulinoma are provided in Supplementary Table 1 and Supplementary Data 1. All of the samples were obtained from subjects not known at the time to be in MEN1 a kindred, although after WES, subject 5967 was found to have a germline *MEN1* mutation.

**Purified normal human beta cells**. RNAseq was performed on purified beta cells from 22 donors of normal human cadaveric islets obtained from three sources. Five {three male, two female; age ($\pm$SEM) = 45.8 $\pm$ 7.0; Body Mass Index (BMI) = 30.6 $\pm$ 2.6} were from islets provided by the NIH/NIDDK-supported Integrated Islet Distribution Program (IIDP) (https://iidp.coh.org/overview.aspx). For these, beta cells were transduced 72 h before harvesting for fluorescence-activated cytometric sorting (FACSAria II) with an adenovirus driven by a RIP1-miniCMV construct that included 177 bases of the hCMV IE-1 promoter ClaI-SpeI fragment ligated to 438 bases of the RIP1 promoter, both upstream of the bright green fluorescent protein ZsGreen (Clontech, Mountain View CA)[25]. The beta cell fraction was confirmed to be >92% pure by immunolabeling of sorted cells with insulin, by quantitative reverse transcription PCR (qRT-PCR) and by RNAseq (see below). For ten others, FASTQ files from normal human cadaveric FACS-sorted beta cells labeled using Newport Green were generously provided by Nica et al.[24]. For the remaining seven, FASTQ files were obtained from Blodgett et al.[23].

**DNA sequencing**. Shearing of 0.5–1 μg genomic DNA to a mean of 200–300 bp fragments was performed using the Covaris E210 focused acoustic energy system (Covaris, Woburn, MA). Whole-genome libraries were prepared using either the NEBNext DNA Library Prep kit (New England Biolabs, Ipswich, MA) or KAPA Hyper Prep kit (Kapa Biosystems, Wilmington, MA) according to the manufacturer's protocol. Illumina compatible paired-end adapters were used, and the adapter-ligated DNA fragments were amplified by ligation-mediated PCR (KAPA Biosystems, Wilmington, MA) using a reverse PCR primer containing a six nucleotide barcode that allowed for multiple samples to be pooled and sequenced in the same run. The library was enriched for exonic sequences with the SeqCap EZ Human Exome Library v3.0 capture system (Roche NimbleGen, Madison, WI). The libraries were then sequenced with a 100 bp paired-end protocol on the Illumina HiSeq 2500 according to standard manufacturer's protocol (Illumina, San Diego, CA).

**Variant calling**. Variant calling and filtering were carried out as described[9, 10]. For each individual insulinoma and blood sample, FASTQ files from all available WES runs were combined into a patient-specific "cohort" and run through an in-house pipeline[9] to yield BAM and VCF files with germline and somatic variant calls (SNVs and small indels). Briefly, this in-house pipeline implements Genome Analysis Toolkit (GATK)[58] version 2.7.2b best practices for alignment, base quality recalibration, variant calling (using HaplotypeCaller), and variant quality score recalibration (VQSR)[59, 60]. VQSR was set to 99.5% sensitivity. Read pairs whose 5′ coordinates were identical were marked (except for one read pair) as duplicates by the Picard software (http://broadinstitute.github.io/picard) and were not used for variant calling, per the above best practices, to ensure that evidence for each variant was derived from distinct DNA molecules, thus avoiding over-counting possibly over-amplified or over-sampled DNA. A GATK genomic interval list was created from the design file from the WES hybridization-capture kit manufacturer;

sequencing depth (Supplementary Data 1) was computed only within these genomic intervals, whereas variant calling was done within these intervals, padded by 100 nt padding on both sides. For somatic variant calling, MuTect[61] (version 1.1.6-10b1ba92, HC + PON mode with default settings, using COSMIC[62] version 65, dbSNP[63] version 137, and using variant calls from patient-matched normal control as the "panel of normals" setting) and Varscan2[64] (version 2.3.5, with flags --tumor-purity 0.7 and --min-var-freq 0.07) were used.

All variant calls were annotated with SnpEff v3.4i[65] (using the Ensembl[66] version 74/GRCh37 resource bundle) and loaded into a custom MySQL (Percona MySQL Server Community Edition 5.6.14-rel62.0.483.rhel6) database schema using in-house scripts, where they were filtered as follows: only variants annotated as altering the amino-acid sequence (missense, nonsense, affecting a canonical splice site, indel in coding sequence) were retained for interpretation; however, the full set of variants was used for routine post-sequencing QC described below. Somatic calls whose population allele frequency in either ESP5400 (http://evs.gs.washington.edu/EVS/)[67] or 1000Genomes[68] exceeded 2% were discarded on the presumption that they are any combination of: contamination, a variant present but missed in normal sample, a low-level artifact, or could not be pathogenic because it was too common in general population. All SNV and indel calls were manually reviewed in IGV and the UCSC Genome Browser[69] to inspect supporting alignment quality in the BAM files and alignability of the genomic region in the hg19 human genome assembly, paying attention to whether a variant call was located in a short tandem repeat or a low-complexity sequence region[70], a region with self-homology/duplication in the reference genome, or a region of low alignability according to the GEM track from ENCODE/CRG[71]. Uncertain calls were manually rejected at this step.

Approximately 70% of the called SNVs had a ref allele of G/C, higher than the GC content of coding regions captured by the WES library capture/enrichment kit. To exclude sequence bias induced by higher GC content, both normal blood and insulinoma from ten randomly sampled subjects were investigated for GC content in the regions with WES coverage above 30×: the GC content ranged between 43–47%. The GC content was 40.4% for the coding regions of all the called genes with SNVs, identical to the GC content of the coding regions of the predicted key drivers (also 40.4%), and very close to the GC content of the coding regions of all the genes in the EpiFactors database (41.9%). Thus, the GC content in regions with reasonable WES coverage and the coding regions of the genes with identified SNVs was comparable. It is unlikely that sequencing bias drives the reported functional enrichment of the predicted key drivers or enrichment in EpiFactor genes. Notably, since mutations are under selection in the context of tumorigenesis, it is anticipated that G/C frequency in our SNVs differs from the G/C content in the WES target region: it is likely these SNVs with higher G/C frequency provide a survival advantage to tumor cells. In addition, it is possible that mutational signatures might also play a role: some mutational signatures simply have much higher G/C ref allele frequency (http://cancer.sanger.ac.uk/cosmic/signatures, for example signature 1–4, 6, 7, etc.), potentially contributing to the insulinoma mutational driver signature landscape.

**Indel force calling**. To further reduce the indel FDR caused by any systematic artifact, we used UnifiedGenotyper (version 2.7)[58] to force-call all indel variants that passed manual review across the entire cohort in order to systematically collect ref/alt allele read depths in all blood/tumor samples. Any indel variant with alt read evidence from other normal blood samples was also removed from the final indel variant list reported.

**Copy number alterations**. Somatic copy number alterations were identified using saasCNV[16]. This algorithm uses heterozygous SNV calls from the normal control sample, and the change in their sequencing depth and allelic fraction in the tumor vs. the normal sample, to identify tumor-specific copy number change events. Joint circular binary segmentation is performed on the two signals: the log-ratio of depth and the log-ratio of mirrored allelic fractions in tumor vs. normal. Identified segments are then classified into one of these categories: loss, cnLOH, gain, normal (no change), or undecided. Because our GATK variant calling workflow combines normal and tumor as a cohort, heterozygous variants called in the normal sample are implicitly force-called by HaplotypeCaller in the tumor sample, yielding their depth-by-allele in both normal and tumor in the output VCF. Variant calls passing VQSR, having a 0/1 genotype in normal, having a mapping quality > 30, and having non-zero depth in tumor, were input into saasCNV. The saasCNV segments classified as loss, gain, and cnLOH CNVs were input to R package RCircos[72] to generate a Circos plot for visualization of recurrent CNV regions. Finally the statistics for significant recurrent CNV regions was analyzed by GISTIC2.0 using the segmentation output of saasCNV with log2ratio values[17]. GISTIC2.0 analysis was performed using default parameters at $q$-value threshold of 0.1.

**Targeted DNA methylation analysis**. Two sets of DNA (300 ng) from four sorted adult beta cell samples was utilized. Three-hundred nanogram DNA was obtained from one sample, and three other sorted beta cell DNA preparations were pooled to achieve the 300 ng minimum. Three-hundred nanogram DNA was prepared from ten insulinomas, of which six were in the original series (4973, 5329, 5333, 5863,

6066, 6107) and four were new samples (5322, 5596, 5597, 5326). Probes were designed to capture the region of chromosome 11 spanning 1.35 Mbp from position 1,850,000 to 3,200,000 within bands p15.5-15.4, (GRCh37/hg19 Assembly Feb 2009) according to the SeqCap Epi Enrichment System (Roche NimbleGen, Roche Sequencing & Life Sciences, Indianapolis, IN, USA) for hybridization-based targeted enrichment of bisulfite-treated DNA. Pre-capture libraries were prepared using the Kapa Hyper Prep kit, PCR-free version (Roche Sequencing & Life Sciences). The 12 samples were barcoded and multiplex-sequenced in a single run and the reads run through a customary DNA methylation pipeline for generating methylation calls at every CpG dinucleotide. Sequencing was performed at the Epigenomics Core Facility of Weill Cornell Medicine. Briefly, 300 ng of genomic DNA were sonicated using a Covaris S220 (Covaris, Woburn, MA, USA) to approximately 180–220 bp fragments. End-repair and A-tailing was performed in a single reaction, followed by ligation of methylated indexed adaptors provided in the Roche SeqCap Epi Enrichment kit. Products were cleaned using Agencourt AMPure XP beads (Beckman Coulter, Indianapolis, IN, USA). Bisulfite conversion was carried out at 54 °C for 1 h using the Zymo EZ DNA Lightning kit (Zymo Research, Irvine, CA, USA), followed by 12-cycles of ligation-mediated PCR amplification performed with HiFi HotSart Uracil + polymerase (Roche Sequencing & Life Sciences). Multiplex hybridization was performed by using 1 µg of bisulfite converted libraries, obtained by pooling 250 ng from each of four libraries, and hybridizing to the custom SeqCap Epi Choice probe pool at 42 °C for 72 h. Hybridized products were purified by capture with Capture Beads and PCR amplified for 15 cycles to create the final libraries for sequencing. Final yields were quantified in a Qubit 2.0 Fluorometer (Life Technologies, Grand Island, NY, USA), and quality of the library was assessed on a DNA1000 Bioanalyzer chip (Agilent Technologies, Santa Clara, CA, USA). Three post-capture multiplexed libraries, each containing four different indexes, were normalized to 2 nM, pooled, clustered at 10 pM on a V2 paired-end read flow cell and sequenced for 150 cycles on an Illumina MiSeq (illumina, San Diego, CA, USA). Primary processing of sequencing images was done using Illumina's Real Time Analysis software (RTA) as suggested by Illumina. CASAVA 1.8.2 software was then used to demultiplex samples, generate raw reads and respective quality scores. Raw data was quality filtered, adapter trimmed and reads aligned to the bisulfite converted reference human genome (GRCh37/hg19 Assembly Feb 2009—whole-genome alignment approach) and the methylation context for each cytosine determined. Custom scripts[73] were used to compute the percent methylation scores and average conversion rates. Average conversion rates obtained ranged from 99.56 to 99.77%. The percent methylation data across the CpG dinucleotides assessed by the targeted DNA methylation platform were averaged within beta cells and insulinomas and compared with each other. The limited number of beta cell control samples ($n = 2$) prevented the use of statistical tests to validate the findings. However the high degree of correlation between the pooled and individual sample (bivariate Pearson correlation coefficient = 0.943—$p < 0.001$; $r^2 = 0.981$—$p < 0.001$) add a substantial degree of confidence to our analysis.

**Pathway enrichment analysis**. Pathway enrichment analysis for the predicted genomic key driver variants was performed using the ClueGo(v2.1.7)[74] and CluePedia(v1.1.7)[75] plugins in Cytoscape(v. 3.1.0)[76] with the GO database (29.02.2016 download). Pathways with a Bonferroni-corrected p-value are shown with full data in Supplementary Data 4. Pathway enrichment analysis for the co-expression modules from transcriptomic analysis was performed by R package goseq with default parameters[77].

**RNA sequencing: beta cells**. RNA from the five sets of FACS-sorted ZsGreen-positive beta cells was prepared immediately using the RNeasy Micro kit (Qiagen). RNA yields were 300–500 ng on each run, and RNA integrity numbers were between 9.5 and 10.0. Briefly, polyA[+] mRNA from sorted beta cells was purified with oligo dT magnetic beads. The polyA RNA from beta cells was then fragmented in the presence of divalent cations at 94 °C. The fragmented RNA was converted into double stranded complementary DNA (cDNA). After polishing the ends of the cDNA, the 3′ ends were adenylated. Finally, Illumina-supplied universal adapters were ligated to the cDNA fragments. The adaptor ligated DNA was size selected to get an average of 250 bp insert size using AmpPure beads, and amplified by 15 cycle PCR. The PCR DNA was then purified using AmpPure beads to get the final seq library ready for sequencing. The insert size and DNA concentration of the seq library was determined on Agilent Bioanalyzer and Qubit, respectively. A pool of ten barcoded RNA seq libraries was layered on two of the eight lanes of the Illumina flow cell at appropriate concentration and bridge amplified to yield ~ 25–35 million raw clusters. The DNA reads on the flow cell were then sequenced on HiSEq 2000 using a 100 bp paired end recipe.

**RNA sequencing: insulinomas**. RNA from frozen insulinoma tissue was prepared using the RNeasy Mini Kit (Qiagen), and fragmented and reverse transcribed as above. One microgram of total RNA was used for the preparation of the sequencing libraries using the RNA Tru Seq Kit (Illumina Cat #1004814). Ribosomal RNA was depleted from total RNA using the Ribozero kit (Invitrogen) to enrich polyadenylated coding RNA and non-coding RNA. The Ribozero RNA-Seq

libraries from insulinomas were sequenced on the Illumina HiSeq 2500 platform using 100 bp paired end protocol following manufacturer's procedure. Base calling from Images and fluorescence intensities of the reads was done in situ on the HiSeq 2500 computer using Illumina software. Various QC parameters such as intensities of individual bases, visual and graphic focus quality of the images, sequence quality measured in terms of colored graphic representation of Q30 values (which is a measure of errors per thousand base), and error rates at 35 and 75 cycles of sequencing, were monitored periodically to assess the quality of an ongoing run.

**RNA-seq alignment and feature quantitation**. Genomic alignment of the paired end RNA-seq reads was performed using STAR[78]. Default parameters for STAR were used, as were those for the quantitation of aligned reads to GRCh37.75 gene features via featureCounts, such that read pairs were counted instead of individual reads[79]. Multimapping reads were flagged and discarded. For splicing inference, the gene annotation was flattened into "exonic" counting bins such that exons with variable lengths across different isoforms were split into multiple bins, including ones of unit length, as in DEXSeq[80]. Reads that overlap multiple counting bins for the same gene were counted for each bin, so when quantitating at the exonic level with featureCount, the -O flag was thrown.

**Differential expression analysis**. DE at both the gene and exonic level, was carried out by first computing feature-wise weights for variance-stabilized counts based on their global mean-variance trend via the voom-transformation[81], and then propagating these feature-wise weights through the limma pipeline to account for the heteroscedastic distribution of count data[82]. A straightforward linear model which accounts for the contrasts of interest, known experimental covariates, and feature-feature correlations between technical replicates, was then fit via generalized least squares and weighted least squares for genes and exons, respectively. The null hypothesis of equal expression was thus tested and adjusted for multiple testing[83], rejection being called at a FDR ≤ 1%.

Given the linear model at the exonic level, differences in exonic retention were tested as to whether exonic log-fold changes in the fit differed for the same gene for contrasts of interest using the diffSplice function within limma. Two tests to quantify evidence of differential exonic usage at the gene-level were used: a F-test for log-fold changes; and the conversion of a series of exonic t-tests into a genewise test using the method of Simes[84]. Relative exonic usage across genes was computed via aggregation of $\log 2(2^x / \Sigma_{i = 1..N}(2^x))$, where $N$ is the number of exons in that particular transcript, and $x$ is exonic expression in units of log2(cpm), at the gene level.

**Allele-specific expression analysis**. ASE was performed by calling heterozygous SNPs from the RNA-seq data and evaluating, for calls which were identified in at least one pair of control beta cells and insulinoma samples, the statistical significance of the reference allele bias with the quasi binomial test against an empirically determined reference fraction. The RNA-seq variant calls were carried out via the GATK variant calling beta practices for RNA-seq, and requiring that each site be supported by at least 30 reads. Calls made in high repeat, low mappability, low complexity regions[85] were removed, as were A/G (T/C) calls in order to recover an empirical null reference allele fraction of ~ 0.54, reflecting the well-known bias toward the reference allele, and to eliminate a likely RNA-editing confounding signal. The resultant SNP coverage was genomically uniform, and the distribution of allelic fractions was approximately normal, with that of beta cell controls slightly more skewed than insulinoma. All features were annotated by SnpEff 3.0 using dbSNP version 138 and GRCh38, Ensembl version 78 and visualized using ggbio, GRanges and ggplot2 within R[86, 87].

**Co-expression analysis**. A signed WGCNA algorithm[26, 88] was used to build co-expression networks, with the log2 of cpm values (after effective library size normalization) as input.

**Bisque4 module membership p-value (MMP)**. For Fig. 5d, the Spearman correlation was calculated among all the genes in the co-expression network and the bisque4 module eigengene (ME). ME was defined as the first principle component of all the genes expressed in a module. The bisque4 MMP was defined as the significance of the Spearman correlation, and the bisque4-associated genes were defined as genes having an MMP < 0.01.

**Histone mark gene set enrichment analysis**. Each of the genes tested in Fig. 3a were ranked by their DE status (insulinoma vs. beta cell) such that genes with logFC ≥ 0 (upregulated in insulinomas) were ranked in an ascending order, followed by genes with logFC ≤ 0 (downregulated in insulinomas) ranked in descending order. Thus in the final gene ranking order, genes with most significant FDR were placed at the end of the x-axis, with each line representing a single gene. Next, we compared these genes tested in DE analysis to published histone mark signatures from prior reports[27–29]. Each histone mark signature was plotted in a single track, with genes in the same order corresponding to the gene order based on DE status as described earlier. In each track, genes with positive histone mark were plotted as black lines. The visualization curves within each histone mark track were

**Table 1 Primer sequences**

|        | Forward primer sequence | Reverse primer sequence |
|--------|-------------------------|-------------------------|
| SLC16A1 | GACCTTGCCGAGCACTTTGA | CCAGGCCGAGGTAGAGATG |
| PDGFRA | TTTTTGTGACGGTCTTGGAAGT | TGTCTGAGTGTGGTTGTAATAGC |
| CXCL12 | ATTCTCAACACTCCAAACTGTGC | ACTTTAGCTTCGGGTCAATGC |
| CDKN1C | GCGGCGATCAAGAAGCTGT | GCTTGGCGAAGAAATCGGAGA |
| LMO4 | GGACAGTCGATTCCTGCGAG | TGTAGTGAAACCGATCTCCCG |
| SLC2A4 | TGGGCGGCATGATTTCCTC | GCCAGGACATTGTTGACCAG |
| IGFBP4 | CATCGTCCTTCCTCTCAAGC | GTCTTCCTTTGACCCCCTTC |
| HK3 | GGACAGGAGCACCCTCATTTC | CCTCCGAATGGCATCTCTCAG |
| PDX1 | ACCAAAGCTCACGCGTGGAAA | TGATGTGTCTCTCGGTCAAGTT |
| NKX6.1 | ACACGAGACCCACTTTTTCCG | TGCTGGACTTGTGCTTCTTCAAC |
| ACTIN | CATGTACGTTGCTATCCAGGC | CTCCTTAATGTCACGCACGAT |
| MAFA | GAGCGGCTACCAGCATCAC | CTCTGGAGTTGGCACTTCTCG |
| KDM5B | CCATAGCCGAGCAGACTGG | GGATACGTGGCGTAAAATGAAGT |
| KDM6A | GGACATGCTGTGTCACATCCT | CTCCTGTTGGTCTCATTTGGTG |
| KDM6B | CTCAACTTGGGCCTCTTCTC | GCCTGTCAGATCCCAGTTCT |
| SLC2A1 | GGCCAAGAGTGTGCTAAAGAA | ACAGCGTTGATGCCAGACAG |
| SLC2A2 | GCTGCTCAACTAATCACCATGC | TGGTCCCAATTTTGAAAACCCC |

plotted as follows: for each histone mark track, each gene was given a binary value (1 for genes with positive histone mark, and 0 for genes with negative histone mark). A sliding window of 201 genes was defined for each gene, including 100 flanking genes on the left and 100 flanking genes on the right, according to the gene-ranking order (increasing or decreasing expression differences between insulinoma vs. beta cells). This was employed to calculate the average gene set scores for each sliding window. These average scores were used to plot the visualization curve for each track.

One-sided Fisher's exact test was performed between genes upregulated in insulinoma (when compared with normal beta cells) and each histone mark signature, with all the protein-coding genes tested in the differential expression analysis between insulinoma and normal beta cells described above as background set.

For the enrichment of histone mark signatures in insulinoma co-expression network (Fig. 3b), a two-sided Fisher's exact test was performed between each histone mark signature and each co-expression module, and FDR was calculated based on all tests. FDR values were then transformed into FDR scores using the following definition: $-\log2(FDR)$ for over-enrichment (odds ratio from fisher exact test >1) and $\log2(FDR)$ for under-enrichment (odds ratio from fisher exact test <1). All insignificant FDR scores (when FDR > 0.05) were converted to a value of 0. The maximum absolute value of FDR score value was set at 50. Finally, the FDR score matrix was used to create the enrichment heat-map shown.

**Adenovirus (Ad) silencing and overexpression**. Ad.shRNAs directed against human CDKN1C, CDKN2A, MEN1, KDM6A, KDM6B, and KDM5B were prepared using the Block-It RNAi kit (Life Technologies) targeting sequence CDKN1C (ATTCTGCACGAGAAGGTACAC), CDKN2A (CGAATAGTTACGGTCGGAG), MEN1 (GATCTACAAGGAGTTCTTTGA), KDM6A (GCAGATACATGGTGT TCAATA)), KDM6B (GCATCTATCTGGAGAGCAAAC), and KDM5B (GCCATCTCCTGTTCTTGTAAA), respectively, driven by the U6 promoter. For overexpression, Ad.EZH2, Ad.CCND1, Ad.YY1, mut-Ad.YY1(T372R) were prepared using the pAd-CMV-V5-DEST Gateway recombination system (Life Technologies)[4, 41]. cDNAs for EZH2 (Harvard Plasmid Library), CCND1 (Addgene), wild-type and mutant YY1 (Harvard Plasmid Library) were cloned into the Gateway pENTR vector. Adenoviruses were packaged and produced in HEK-293A cells. Titers were determined by plaque assay (PFU). Prior to transduction, islets were dispersed using Accutase Cell Detachment Solution (Innovative Cell Technologies, San Diego, CA), then transduced with 150 MOI in serum-free RPMI-1640 medium containing 1% penicillin, 1% streptomycin, and 5.5 mM glucose. After 2 h, transduction was terminated by addition of complete medium containing 10% fetal bovine serum, and the islets were cultured for 72–96 h.

**Immunocytochemistry and immunohistochemistry**. Immunocytochemistry was performed using standard methods[4, 41]. Briefly, islets were dispersed, transduced, and plated on cover slips. Insulin (DAKO, Carpinteria, CA) and Ki67 (Fisher Scientific) primary antisera were used to immunolabel beta cells entering the cell cycle. A tissue microarray containing formalin-fixed, paraffin-embedded tissue sections of insulinoma and normal pancreas tissue from the same and different clinical cases were generated by the Biorepository and Pathology CORE at the Icahn School of Medicine at Mount Sinai. Primary antisera against p57 (Cell Signaling, Danvers, MA) and insulin (DAKO, Carpinteria, CA) were used to label

p57 in human beta cells. Secondary antisera were from Invitrogen (Carlsbad, CA). Confocal fluorescent microscopy was performed using an Olympus Fluoview 1000 microscope.

**Chromatin immunoprecipitation assays**. ChIP was performed using the EZ-ChIP Kit (Millipore) according to the manufacturer's protocol. Beta cells were FACS sorted using Ad.ZsGreen as described earlier. Forty-thousand ZsGreen+ (beta) and ZsGreen− (non-beta) cells were collected per experiment for each H3K4me3 and H3K27me3 immunoprecipitation, and 130,000 sorted cells were collected for each KDM6A immunoprecipitation. The primer sets were designed in a previous study[43]. Immunoprecipitated DNA was quantified using ABI 7500 real-time quantitative PCR detection system (Life Technologies). The following antibodies were used: anti-H3K4me3 (Millipore #17-614), anti-H3K27me3 (Millipore #17-622), anti-KDM6A (Abcam #ab84190). Data are presented as ChIP reads normalized relative to input controls, and fold-enrichment divided by respective IgG. Three separate islets preparation were used for each figure shown.

**Gene expression analysis**. Ten nanogram of total RNA from isolated FACS-purified beta cells or insulinoma tissue samples were reverse transcribed using the SuperScript III First-Strand Synthesis SuperMix kit (Life Technologies). Gene expression was analyzed by real-time PCR performed using an ABI 7500 system. For RT-PCR, all data are expressed as the mean ± SEM. Results were accepted as statistically significant at $p < 0.05$, as determined using two-tailed Mann–Whitney test. Primer sequences used are listed in Table 1.

**Data availability**. DNAseq, RNAseq, and CpG Bisulfite seq raw data files are available through dbGAP (https://www.ncbi.nlm.nih.gov/projects/gap/cgi-bin/about.html) under accession number: phs001422.v1.p1 and The Catalogue of Somatic Mutations in Cancer (COSMIC) via ID number: COSP44132.

Processed data are also available on our Insulinoma Genomic Portal at http://insulinoma.genomicportal.org

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

## Acknowledgements

We wish to thank the following people and organizations for their support: The Foundation for Diabetes Research, the Murray-Heilig Fund in Molecular Medicine at the University of Connecticut; the NIDDK Islet Integrated Distribution Program (IIDP); Dr Tatsuya Kin at the University of Edmonton and Dr Patrick MacDonald at the Alberta Diabetes Institute; The Biorepository and Pathology Core, the Flow Cytometry Center for Research Excellence, and the computational resources and staff expertize provided by the Department of Scientific Computing at the Icahn School of Medicine at Mount Sinai; The Human Islet and Adenoviral Core (HIAC) of the Einstein-Sinai Diabetes Research Center (DRC); The Epigenomics Core of Weill-Cornell Medicine. We thank Drs Ronald Tamler, Erika Villanueva for helping to obtain insulinomas; Drs Yanan Cao and Guang Ning of the Shanghai Key Laboratory for sharing data; Mr Justin Bellizzi for technical assistance; and Drs Alexandra Nica, Philippe Halban, Manolis Dermatzakis, and Drs David Blodgett and David Harlan for sharing their normal beta cell transcriptome data with us. We thank Drs Adolfo Garcia-Ocaña and Rupangi Vasavada for continued discussion of these findings, Dr Jordi Ochando for help with human beta cell flow cytometry, Dr Christopher Newgard for sharing the RIP1-mini-CMV enhancer promoter for construction of the RIP1-ZsGreen adenovirus, and Ms. Aye Moe and Drs Wei-Yi Cheng, Ke Hao and Antonio di Narzo for bioinformatic support. Finally, we apologize to the authors whose important work we were unable to cite because of reference number limitations. This work was supported by NIDDK grants UC4 DK104211 and P-30 DK 020541.

## Author contributions

W.B.I., S.K.L., A.A., M.S., H.C., J.F., G.F.-R., A.F.S. collected and contributed insulinomas and/or DNA or RNA and/or clinical information. M.D., A.F.S., A.B., maintained the Insulinoma Biorepository. H.W., Y.Ka., M.M., Y.Ki., R.C., D.K.S., B.R., A.V.U., L.L. contributed to DNAseq and its analysis. H.W., C.A., B.L. performed RNAseq analysis. E.K., P.W., performed ChIP studies. H.W., C.A., E.E.S., B.L., J.F., A.V.U., B.R., L.L. each led a major aspect of the bioinformatic analysis. Y.A. built the Human Insulinoma Portal. A.B., P.W., S.S., E.K., L.A.W., K.K.T., T.C.B., D.K.S. prepared adenoviral vectors, and/or performed validation experiments in human islets and/or insulinomas. H.W., A.B., C.A., A.V.U., B.L., A.A., J.F., E.E.S., A.F.S. wrote the manuscript. A.F.S., W.B.I., S.K.L., M.S. conceived of the study.

## Additional information

**Competing interests:** The authors declare no competing financial interests.

