## [Peer Review File · Nature Communications]

Reviewers' Comments:

Reviewer #1:

Remarks to the Author:

Comments for the Authors:

This is an important study providing insights into the potential mechanisms of human beta cell proliferation, and with future data-mining opportunities for other studies. The authors have addressed my previous questions and concerns satisfactorily. Here are some additional comments for the new text in the current version (page and line numbers are from the marked-up version of the manuscript text):

- 1) Page-3, 9th line from the bottom: Including MEN1 as a "recurrently" mutated gene is not correct here. Only one tumor showed a somatic MEN1 gene mutation.
- 2) Page-3 (9th line from the top) and page-4 (2nd line from the top): "functionally related to MEN1". To which function of MEN1?
- 3) Page-4, first paragraph: For CNVs, the locus of MEN1 (11q13.1) and CDKN1C (11p15, incorrectly written here as 11q15) is given but the locus of EZH2 is not given (7q36.1). This region should be included as amplified in Fig. 1f.
- 4) Page-6, 7th line from the bottom: Instead of "humans", please include "in human MEN1-associated insulinoma". Note that MEN1 gene mutation is a well-known cause of insulinoma in the human MEN1 syndrome, but MEN1 gene mutation is rarely seen in human non-MEN1 insulinomas.
- 5) Page-11, end of the first paragraph: "Mutation of MEN1 alone cannot cause insulinoma". But MEN1 gene mutation was seen in only one of 25 (37?) sporadic (non-MEN1) insulinoma.
- 6) Page-11, 6-10th line from the bottom: Were any genetic or transcriptomic differences observed between the benign insulinomas and the 2 malignant insulinomas included in this study?
- 7) Page-12, 2nd paragraph: CTIIA alternative splicing events. Does this predict loss of functional CTIIA?
- 8) Fig. 2b and Suppl. Fig. 5: GCG and PPY can be explained in the legend as glucagon and pancreatic polypeptide Y. In the legend of Suppl. Fig. 5, "MEN1" not MEN, and also in the graphs of Suppl. Fig. 8, 9 and 10.
- 9) Fig. 5g, at the end of the legend: Note that EZH2 and shCDKN1C are not shown in the Figure.
- 10) Fig. 5g and Suppl. Fig. 5: Was it possible to look at only those cells that were transfected? In other words, was it possible to factor-in transfection/transduction efficiency of the specific islet cell types on the Y-axis? This may affect (increase? or decrease?) the percent Ki67 positive cell number.
- 11) Suppl. Fig. 5 and page-9 bottom: Effect of silencing CDKN1C in non-beta islet cells appears to be irrelevant because on page-10, line 8/9, it is stated that CDKN1C is only expressed in beta cells in human islets.
- 12) Fig. 6a: Black = mutations? Most of them are missense variants. Therefore, should black indicate "missense variants" of "unknown significance"? Also, "red" is not mentioned in the figure legend.

Reviewer #2:

Remarks to the Author:

The authors have addressed all my concerns.

Reviewer #3:

Remarks to the Author:

The authors have significantly improved the original version of the manuscript through addition of relevant new data sets and thorough editorial changes. My prior comments have been addressed adequately. Overall, the findings presented in the manuscript provide compelling, comprehensive, and state of the art insights into the genetic state of human insulinoma.

Reviewer #4:

Remarks to the Author:

This manuscript has already been fairly extensively reviewed, and in general I agree with the other reviewers' comments. I have a few specific points relating to the analysis of the data presented, which I would like the authors at least to comment on in order to clarify the readers' interpretation of the results.

1) Fold-enrichment of EpiFactors genes in predicted drivers is modest. The developmental signatures seem more markedly enriched, I'm curious as to why these were not mentioned in the manuscript?

2) Related to the enrichment of EpiFactors in predicted drivers, a quick glance at the distribution SNV frequencies affecting a C/G reference allele seems high (70%) even for exomes. Is this similar to the expected frequency across your WES regions with reasonable coverage across the set of insulinomas? Sequencing depth tends to be limited at high-GC and low-GC regions (see Clark et al, 2011 Nat Biotech), therefore if you compare the GC content of the coding regions for the predicted driver genes, are these over-represented in the 'good-GC-content' range? What about EpiFactor genes? It could be that the amplification/sequencing bias is driving the reported functional enrichments, therefore I consider it important to rule this out.

3) You state that insulinomas feature hypermethylation 'in a few selected areas containing sequences known to have important regulatory functions in imprinting regulation', but it seems there is only one specifically mentioned (KvDMR1) and shown in Fig4. Can you really justify this general statement? Also, it seems interesting that the differential methylation around CDKN1C appears downstream of the gene (assuming it is transcribed as indicated with the arrow in Fig4). Are there any described regulatory elements in this region? Does this link at all with the KDM6A binding upstream of the gene? I appreciate that the Trithorax-linked mechanisms relate to histone modification and not DNA methylation, but without any link here it seems to me that the DNA methylation data doesn't really add anything clear to the story.

4) Given the conservation of WGCNA modules across insulinomas and normal islet cells, I wonder if looking for co-expression networks within the insulinoma vs beta cell DEGs might point towards a mechanism of wider cell cycle dysregulation in the insulinomas? This might in turn yield more candidate drivers for the altered proliferative phenotype?

5) In description of histone mark geneset enrichment analysis (methods), how are genes defined as 'with positive histone mark' or 'with negative histone mark'? Also, when you say '100 flanking genes on the left/right' does this mean in increasing/decreasing order of insulinoma vs beta cell differential expression, or in genomic co-ordinates? Please clarify in text.

Reviewer #1

“This is an important study providing insights into the potential mechanisms of human beta cell proliferation, and with future data-mining opportunities for other studies. The authors have addressed my previous questions and concerns satisfactorily. Here are some additional comments for the new text in the current version (page and line numbers are from the marked-up version of the manuscript text):”

Response: Thank you once again for your very helpful comments.

1) *Page-3, 9th line from the bottom: “Including MEN1 as a “recurrently” mutated gene is not correct here. Only one tumor showed a somatic MEN1 gene mutation.”*

Response: While we agree that the reviewer is technically correct, we explicitly make this distinction two sentences earlier (“Only two tumors had *MEN1* mutations, one somatic and one germline.”). Since the *MEN1* mutations in both insulinomas are protein changing/disabling, we believe it is reasonable to include *MEN1* mutations among the recurrent mutations for the purposes of this discussion: the parental source of the mutation is less important here than the presence of the mutated gene. If the Reviewer feels strongly, we could add an additional sentence further clarifying this point, but feel that it is clear as written. We of course defer to the Editor on this point. Thank you.

2) *Page-3 (9th line from the top) and page-4 (2nd line from the top): “ ‘functionally related to MEN1’ ”. To which function of MEN1?”*

Response: We have expanded this sentence to add clarity here, as follows: “Thus, surprisingly, despite having largely excluded *MEN1*-associated insulinomas, the strongest recurrent mutational signal nonetheless arose from genes encoding epigenetic modifiers functionally related to *MEN1*, which also encodes an epigenetic modifying enzyme”.

3) *Page-4, first paragraph: “For CNVs, the locus of MEN1 (11q13.1) and CDKN1C (11p15, incorrectly written here as 11q15) is given but the locus of EZH2 is not given (7q36.1). This region should be included as amplified in Fig. 1f.”*

Response: Thank you for catching these errors and omissions. We have corrected 11p15, and added 7q36.1 to paragraph 4 as requested.

Regarding *EZH2*, as shown in the Circos plot, it is detected by the saasCNV3.3 algorithm as having recurrent CNV events. However, for the Gistic plot, it did not meet the q value threshold (<0.1) shown for Fig 1f. Notably, chr7 (containing *EZH2*) appears to be recurrently gained (probably aneuploid) according to the Circos plot in Fig 1e. However, the saasCNV segmentation algorithm (the input into Gistic) may be fragmenting the putative aneuploidy into smaller segments due to noise inherent in the WES data, instead of capturing the entire chromosome as one segment/CNV event. Thus, although the copy gain of *EZH2* does indeed occur in multiple samples (saasCNV; Fig 1b, 1e), the lack of significant support by GISTIC for the *EZH2* locus may be

due to a combination of factors: 1) lack of sensitivity by GISTIC2.0 due to small cohort size; 2) lack of sensitivity by GISTIC2.0 due to noisy segmentation profiles generated from WES data by saasCNV; and, 3) GISTIC2.0 favoring focal segments whereas these gains occur on large segments (mean 34.8 Mb, range 15.6-145.1 Mb). However, note that GISTIC does assign statistically significant copy gain recurrence to most of chr7 (Fig 1f), so it is possible that with a larger sample size, e.g. cohorts of >100 samples on which GISTIC2.0 is generally used, the EZH2 locus and/or more of chr7 may be called significantly recurrent.

We have not revised the text on this point, but believe that *Nature Communications* publishes the Reviewer Comments and Author Responses so this will be available to readers of the manuscript. Of course, we are happy to add this to the Online Supplementary Methods if the Editor and Review prefer. Thank you.

4) Page-6, 7th line from the bottom: *“Instead of ‘humans’, please include ‘in human MEN1-associated insulinoma’. Note that MEN1 gene mutation is a well-known cause of insulinoma in the human MEN1 syndrome, but MEN1 gene mutation is rarely seen in human non-MEN1 insulinomas.”*

Response: We agree and have made this change. Thank you.

5) Page-11, end of the first paragraph: *“ ‘Mutation of MEN1 alone cannot cause insulinoma’. But MEN1 gene mutation was seen in only one of 25 (37?) sporadic (non-MEN1) insulinoma.”*

Response: The point we are trying to make is that patients with insulinomas require “hits” in multiple genes. We have expanded this sentence to make the point more clearly. Thank you.

6) Page-11, 6-10th line from the bottom: *“Were any genetic or transcriptomic differences observed between the benign insulinomas and the 2 malignant insulinomas included in this study?”*

Response: This is a good question. At this point, there are a number of interesting differences, but nothing that we can be comfortable about in statistical sense. We are actively expanding the series and hope to be able to answer this in with statistical confidence in the future.

7) Page-12, 2nd paragraph: *“CTIIA alternative splicing events. Does this predict loss of functional CTIIA?”*

Response: This is a very interesting question. The short answer is that we do not know, and could not address/biologically pursue every interesting lead in this first paper, so elected to focus our biological understanding/mechanism/validation efforts on beta cell replication, the principal purpose of the study.

The longer answer is that we have no bioinformatic evidence that the CIITA splicing abnormalities have functional consequences related to impaired MHC-II signaling. As the Reviewer may be aware, multiple CIITA isoforms have been observed, and can be divided into two broad classes (“short” with 10 or 11 exons and “long” with 19 or 17 exons). Here, it appears that the short isoform is preferentially expressed in insulinomas as compared to normal control beta cells. The precise functional role of each class of isoform is unknown, but generally CIITA is known as a prominent positive regulatory of MHC-II expression (*Int. Immunol* <https://doi.org/10.1093/intimm/dfx060>, PMID 12147620). Overall, since we can only speculate, we have not added new text to the manuscript, but would be happy to add thoughts along these lines if the Reviewer and Editor feel it is important to do so. Thank you.

8) Fig. 2b and Suppl. Fig. 5: *“GCG and PPY can be explained in the legend as glucagon and pancreatic polypeptide Y. In the legend of Suppl. Fig. 5, ‘MEN1’ not MEN, and also in the graphs of Suppl. Fig. 8, 9 and 10.”*

Response: Thank you for catching these. We have made these changes.

9) *Fig. 5g, at the end of the legend: "Note that EZH2 and shCDKN1C are not shown in the Figure."*

Response: Thanks for catching this. The Ad.EZH2 and Ad.shCDKN1C insulin-Ki67 photomicrographs were shown in the last version. We "shrunk" the inset in this Figure in the revision (which had included examples of Ad.EZH2 and Ad.shCDKN1C on Ki67) in order to include the new GSK J1 and ChIP data in new panels 5.h-j. We have revised the Legend to reflect this as suggested.

10) *Fig. 5g and Suppl. Fig. 5: "Was it possible to look at only those cells that were transfected? In other words, was it possible to factor-in transfection/transduction efficiency of the specific islet cell types on the Y-axis? This may affect (increase? or decrease?) the percent Ki67 positive cell number."*

Response: This is a good question. The efficacy of adenoviral transduction in dispersed islet cells is approximately 80-90%, so it is reasonable to assume that the majority of cells were transduced.

11) *Suppl. Fig. 5 and page-9 bottom: "Effect of silencing CDKN1C in non-beta islet cells appears to be irrelevant because on page-10, line 8/9, it is stated that CDKN1C is only expressed in beta cells in human islets."*

Response: The Reviewer is correct. The alpha, delta and PP cells data in Suppl. Fig 5 provide an additional negative control for the mitogenic effects of Ad.shCDKN1C on beta cells, illustrating that like the other negative control (Ad.shLacZ), Ad.shCDKN1C has no effects on cells that do not express CDKN1C. Thank you.

12) *Fig. 6a: Black = mutations? Most of them are missense variants. Therefore, should black indicate 'missense variants' of 'unknown significance'? Also, 'red' is not mentioned in the figure legend.*

Response: Thank you for catching these points. We have corrected them. We have left the "black" definition as "missense variants", because the YY1 mutation is believed to be an activating mutation, and the other SNVs/MNVs and Indels are largely protein structure altering.

Reviewer #2

"The authors have addressed all my concerns."

Response: Thank you for your help and support in the review process!

Reviewer #3

"The authors have significantly improved the original version of the manuscript through addition of relevant new data sets and thorough editorial changes. My prior comments have been addressed adequately. Overall, the findings presented in the manuscript provide compelling, comprehensive, and state of the art insights into the genetic state of human insulinoma."

Response: Thank you also for your helpful comments and support during the review process!

Reviewer #4

“This manuscript has already been fairly extensively reviewed, and in general I agree with the other reviewers’ comments. I have a few specific points relating to the analysis of the data presented, which I would like the authors at least to comment on in order to clarify the readers’ interpretation of the results.”

Response: Thank you for your positive feedback and helpful suggestions.

1) *“Fold-enrichment of EpiFactors genes in predicted drivers is modest. The developmental signatures seem more markedly enriched, I’m curious as to why these were not mentioned in the manuscript?”*

Response: Thank you for this comment. We agree that while the EpiFactors enrichment in the 92 key drivers was more modest than the GO enrichment, the enrichment in the recurrent genes (which are also considered key drivers) was very high in EpiFactors (Fold Enrichment = 16.7, $p=2.8 \times 10^{-6}$), suggesting agreement between both types of analyses. We cited both types of enrichment in the main text (the paragraph spanning pages 3 and 4) to illustrate this. However, we agree that development pathways are also enriched, likely reflecting the fact the major role of chromatin-modifying enzymes in development and differentiation. We have therefore added a final sentence to this paragraph on page 4 with the Reviewer’s comment in mind.

2) *“Related to the enrichment of EpiFactors in predicted drivers, a quick glance at the distribution SNV frequencies affecting a C/G reference allele seems high (70%) even for exomes. Is this similar to the expected frequency across your WES regions with reasonable coverage across the set of insulinomas? Sequencing depth tends to be limited at high-GC and low-GC regions (see Clark et al, 2011 Nat Biotech), therefore if you compare the GC content of the coding regions for the predicted driver genes, are these over-represented in the ‘good-GC-content’ range? What about EpiFactor genes? It could be that the amplification/sequencing bias is driving the reported functional enrichments, therefore I consider it important to rule this out.”*

Response: Thanks for this interesting and important question. We have performed additional analyses and are reassured that the findings relate to insulinoma biology rather than technical sequencing issues.

First, as the Reviewer suggested, we have randomly sampled 10 patients (in both normal blood and tumor tissues) for their GC content in the regions with WES coverage above 30x, and the GC content ranges between 43 - 47%. The GC content of the coding regions of all the genes with SNVs we called is 40.4%, the GC content of the coding regions of the predicted key drivers is also 40.4%, and the GC content of all the genes in the EpiFactors database is 41.9%. Thus, it doesn’t appear that a large GC content difference exists between regions with reasonable WES coverage and the coding regions of the genes where we identified SNVs, making it unlikely that sequencing bias is driving the reported functional enrichment of the predicted key drivers or enrichment in EpiFactor genes.

Second, mutations are under selection in the context of tumorigenesis. Thus, it is normal that G/C frequency in our SNVs differs from the G/C frequency in our WES target region: it is likely the SNVs with higher G/C frequency provide a survival advantage to tumor cells in some way. In addition, it is possible that mutational signatures might also play a role: some mutational signatures simply have much higher G/C ref allele frequency (<http://cancer.sanger.ac.uk/cosmic/signatures>, for example signature 1-4, 6, 7, etc.). These might possibly be the driving signatures for insulinoma mutational landscape. This provides an additional explanation as to the observation that our SNVs have a GC reference allele frequency of 70%.

We have not revised the text on this point, but believe that *Nature Communications* publishes the Reviewer Comments and Author Responses so this will be available to readers of the manuscript. Of course, we are happy to add this to the Online Supplementary Methods if the Editor and Review prefer. Thank you.

3) Comment 3 has three separate components, all of which are excellent questions. We are grateful for the opportunity to address them, but simply have no space in the main text or figure legends to address these in detail: we are already at the maximum 5000 words permitted. Thus, we respond to the three sub-comments/questions below, and again, presume and hope that the Editor will publish the Reviewer comments and our responses, as per *Nature Communications* policy, so that interested readers can explore these questions more deeply. Overall, however, the evidence in the manuscript and below is overwhelming that there will be imprinting and 3-D structural and chromatin looping abnormalities in the 11p15 region, and these should be documented with CTCF ChIPseq, and chromatin capture techniques. We agree that these are important future studies, but are beyond the scope of the current study, and hope the Reviewer will indulge us on this point. We have added a brief sentence/phrase to the very top of page 8 with these thoughts in mind. Thank you.

3a) *“You state that insulinomas feature hypermethylation ‘in a few selected areas containing sequences known to have important regulatory functions in imprinting regulation’, but it seems there is only one specifically mentioned (KvDMR1) and shown in Fig4. Can you really justify this general statement? Also, it seems interesting that the differential methylation around CDKN1C appears downstream of the gene (assuming it is transcribed as indicated with the arrow in Fig4). Are there any described regulatory elements in this region?”*

Response: The first of the three major sub-regions in Figure 4f, g and h contains two of the five promoters (P0-P4) that control the tissue- and time-specific expression of *IGF2* (*Hum Mol Genet*: 15 (8): 1259-69; 2006, PMID 16531418). The differential activation of these promoters is deeply intertwined with the mono- vs bi-allelic expression (imprinting status) of the *IGF2*, *INS* and *INS-IGF2* readthrough transcripts (*Hum Mol Genet*: 15 (8): 1259-69; 2006 PMID16531418). It is clear that the P1 promoter is located in a hypermethylated area in this subregion, and that the P0 promoter and the DMR0 ICR are in a hypomethylated area. Thus, it fair to say that within this first sub-region sequences with known important regulatory functions in imprinting regulation (P1 promoter) are abnormally methylated.

The third sub-region (in the same panels described above) carries hypermethylation of sequences with important regulatory functions involved in imprinting regulation. For example, hypermethylation in the third sub-region cover a portions of *CDKN1C* promoters 1 and 2 (captured with our SeqCap Epi Enrichment System design) and both enhancers located upstream and downstream the gene (the whole gene is characterized by a cluster of closely arranged enhancers). The *CDKN1C* promoter and surrounding enhancers have been implicated in the imprinting-associated regulation of this area (Clin Epigenetics 69: eCollection 2016, PMID 27313795).

Finally, we have confirmed that the direction of transcription of *CDKN1C* reported in Fig. 4f is correct as per the UCSC genome browser.

3b) *“Does this link at all with the KDM6A binding upstream of the gene?”*

Response: We agree it is confusing as written, because the directionality in the UCSC genome browser (in Fig 4f-h) and the *CDKN1C* ChIP experiments and diagram (in Fig. 5j) go in opposite directions. To clarify, the third shaded region of hypermethylation in the third panel in Fig. 4f-h overlaps with the “active enhancer” for *CDKN1C* in Fig. 5j. The specific region of KDM6A binding to *CDKN1C* (primer pair 2 in Fig. 5j), which also overlaps with the enhancer, is hypomethylated in insulinomas compared to normal beta cells. Thus, the *CDKN1C* enhancer region displays areas of both hypo- and hyper-methylation, which likely result in altered CTCF binding, DNA looping, insulator function, again, all best assessed by 5-C, 3-C and ATACseq in future studies, as described at the top of page 8. For the current report, we assume this discussion will be published with the manuscript, as alluded to in our response Comment 2 above. In the main manuscript, we have clarified the directionality issue in the Legend to Figure 5. Thank you.

3c) *“I appreciate that the Trithorax-linked mechanisms relate to histone modification and not DNA methylation, but without any link here it seems to me that the DNA methylation data doesn't really add anything clear to the story.”*

Response: Again, we realize that there is a lot in the manuscript, and it is difficult to cover every detail with the clarity they merit. The “big picture” here, as we say in the beginning of the second paragraph of the Discussion on p.11, is that “many epigenetic roads may lead to insulinoma”. The two principal “roads”, both novel, are the recurring TrxG, PRC chromatin modifying enzyme “signatures”, and the striking mutation, CNV, ASE and methylation “signatures” in the 11p15 region, a region all ready well known as an imprinted region that is abnormal in two rare human diseases of beta cell replication. We do not know if, and do not propose that, the two signatures are related, although it is plausible. What we have done in the methylation component of this study is to perform the first and by far the deepest nex-gen CpG sequencing in this region in insulinoma and find it markedly abnormal. While we that we have not shown that methylation abnormalities directly lead to gene expression abnormalities in this region, the question at this point not “whether they exist” but rather what exactly they are. As we note above, this will require CTCF ChIPseq, ATACseq, 5-C/3-C studies which have never been done before in beta cells or insulinoma. Given the requirement for fresh insulinomas and their rarity, this will take time, and again is beyond the scope of this study. We note however, that we have proposed these very studies in a pending NIH application. Again we hope the Reviewer will indulge us on this point.

4) *“Given the conservation of WGCNA modules across insulinomas and normal islet cells, I wonder if looking for co-expression networks within the insulinoma vs beta cell DEGs might point towards a mechanism of wider cell cycle dysregulation in the insulinomas? This might in turn yield more candidate drivers for the altered proliferative phenotype?”*

Response: We thank the Reviewer for this interesting suggestion. From a feasibility standpoint, making a co-expression network based only on DEGs between beta cells and insulinomas can be done. However, as reported previously, we focus on either all expressed genes, or at least to topmost varying genes across the samples used as inputs in the co-expression network, i.e., the insulinomas (Zhang B....Schadt EES et al *CELL*, 2013 PMC 3677161). To assess whether that strategy might point towards wider cell cycle dysregulation in the insulinomas, we investigated which canonical pathways (GO, Reactome etc) are enriched in the DEG list. Among these ~3000 DEGs, however, there was no significant enrichment found for cell cycle pathways. The signal is dominated by extracellular matrix pathways (which were also prominent in Figure 2e, the brown module in the WGCNA). As this canonical pathway enrichment analysis could be considered similar to a co-expression network analysis, it seems unlikely that constructing a WGCNA based solely on the DEGs would be able to point towards a mechanism of wider cell cycle dysregulation in the insulinomas.

At a higher level, we believe this reflects a broader issue in human beta cell proliferation research. Unlike cancer studies in which tumor cell proliferation rates may be very high, in benign insulinomas and in normal neonatal human beta cell proliferation, proliferation rates are very low in the range of 0.5-3%, as described in references 1 and 44. Thus, the lack of cell cycle enrichment in the DEG list alone likely reflects the low proportion (1-3%) of beta cells in active proliferation in our bulk RNAsequencing sample, and their dilution by the 97-99% of beta cells that are not actively replicating.

With this background, we believe it is reasonable to have used our previously published approach (Zhang B....Schadt EES et al *CELL*, 2013 PMC 3677161), generating co-expression networks using all the expressed genes and then highlighting modules with perturbed function through the use of differentially expressed genes from contrasts of interests (e.g. insulinoma vs. beta). Fortunately, this approach was able, in an unbiased, data-driven way, to uncover a module that was significantly enriched for cell cycle genes. It also was effective in revealing co-regulated genes such as *CDKN1C* and *EZH2* that we were able to validate as being important in the cell cycle process.

With the Reviewer’s suggestion in mind, we have emphasized on page 9, final paragraph, that while we chose to focus on and validate a subset of genes associated with the cell-cycle module for their role in cell cycle, other candidates on that list (Supplemental Table 20, in particular) could also be potential key drivers. We provide the list for the greater community to explore as well. Finally, we think this analysis is a good example of the power of the genome-wide co-expression network structure, which is the ability to capture genes that are important, yet escape one’s DEG list, either because of a threshold that was used or because they simply were

not actively being up or down-regulated at the time the sample was collected.

5) *"In description of histone mark geneset enrichment analysis (methods), how are genes defined as 'with positive histone mark' or 'with negative histone mark'? Also, when you say '100 flanking genes on the left/right' does this mean in increasing/decreasing order of insulinoma vs beta cell differential expression, or in genomic co-ordinates? Please clarify in text."*

Response: Yes, this is correct, in that the '100 flanking genes on the left/right' does this mean in increasing/decreasing order of insulinoma vs, beta cell differential expression". We have clarified this in the revised Supplementary Methods on p. 17. Thank you.

Again, we want to thank all four Reviewers for helping us to generate what we believe is an even stronger and clearer manuscript. We hope that all these changes are sufficient to allow acceptance.

Reviewers' Comments:

Reviewer #4:

Remarks to the Author:

The authors have addressed my concerns in their detailed response letter. I appreciate the article word limits, but am a bit disappointed that no mention is made (in either manuscript or supplementary material) to the response to my comment #2. Given there is a strong sequence bias in the observed SNVs, if this is not a technical issue then it could be an important aspect of the pathology (either in terms of exposures or selection) representing a difference between insulinomas and normal beta cells.